# Ubiquitin is directly linked via an ester to protein-conjugated mono-ADP-ribose

Daniel S Bejan [ID][1], Rachel E Lacoursiere [ID][2], Jonathan N Pruneda [ID][1,2,3]✉ & Michael S Cohen [ID][1,3]✉

## Abstract

The prevailing view on post-translational modifications (PTMs) is that a single amino acid is modified with a single PTM at any given time. However, recent work has demonstrated crosstalk between different PTMs, some occurring on the same residue. Such interplay is seen with ADP-ribosylation and ubiquitylation. For example, DELTEX E3 ligases were reported to ubiquitylate a hydroxyl group on free NAD+ and ADP-ribose in vitro, generating a noncanonical ubiquitin ester-linked species. In this report, we show, for the first time, that this dual PTM occurs in cells on mono-ADP-ribosylated (MARylated) PARP10 on Glu/Asp sites to form a MAR ubiquitin ester. We call this process mono-ADP-ribosyl ubiquitylation or MARUbylation. Using chemical and enzymatic treatments, including a newly characterized bacterial deubiquitinase with esterase-specific activity, we discovered that multiple PARPs are MAR-Ubylated and extended with K11-linked polyubiquitin chains when exogenously expressed. Finally, we show that in response to type I interferon stimulation, MARUbylation can occur endogenously on PARP targets. Thus, MARUbylation represents a new dual PTM that broadens our understanding of the function of PARP-mediated ADP-ribosylation in cells.

**Keywords** ADP-ribosylation; PARPs; Ubiquitylation; Post-translational Modifications; Interferon Response
**Subject Category** Post-translational Modifications & Proteolysis

## Introduction

Post-translational modifications (PTMs) impact protein function, localization, stability, and interactions, ultimately governing key cellular processes essential for life. PTMs can range from minor alterations, like adding a methyl group (0.015 kDa), to large changes, such as conjugation of ubiquitin (~8.5 kDa). PTMs provide proteins with a wide range of functional and structural diversity. These modifications can happen individually, at various sites within a protein, or sometimes, multiple modifications can occur at one specific site. Two enzyme families exemplify the latter, namely PARPs and ubiquitin ligases. PARPs, a family of 17 members in humans, transfer ADP-ribose (ADPr) from nicotinamide adenine dinucleotide (NAD+) onto target substrates in a process called ADP-ribosylation. Mono-ADP-ribosylation (MARylation) is the transfer of a single unit of ADPr, whereas poly-ADP-ribosylation (PARylation) involves the transfer of multiple units of ADPr, up to 190 in vitro (Alvarez-Gonzalez and Jacobson, 1987). The PARP members are divided based on their MARylation (PARP3, 4, 6–12, 14–16) or PARylation (PARP1, PARP2, tankyrase 1 (TNKS1), tankyrase 2 (TNKS2)) activity, which can occur on nucleophilic amino acids, including glutamate, aspartate, serine, arginine, and cysteine (Cohen and Chang, 2018). Ubiquitylation (or ubiquitination) is a multi-step process involving ubiquitin (Ub) activation by E1 enzymes, conjugation by E2 enzymes, and finally transfer onto target substrates by E3 Ub ligases. Ub is canonically attached to lysine residues of a target protein via its carboxy-terminal glycine, generating an isopeptide linkage. This monoUb species can then be extended into polyUb chains through seven different lysines (K6, K11, K27, K29, K33, K48, K63) or the amino-terminal methionine (M1) of Ub (Swatek and Komander, 2016).

A growing body of work indicates the existence of crosstalk between ADP-ribosylation and ubiquitylation in both prokaryotes and eukaryotes. Prokaryotes do not encode a canonical Ub system. However, bacterial pathogens can disrupt host ubiquitylation pathways by secreting specialized effector proteins (Maculins et al, 2016; Roberts et al, 2023). For example, *Legionella pneumophila* secretes SdeA, a Ub ligase of the SidE family that ubiquitylates host substrates independent of ATP and canonical E1 and E2 enzymes (Qiu et al, 2016). This is achieved through concerted activities of the mono-ADP-ribosyltransferase (mART) and phosphodiesterase (PDE) domains in SdeA. First, SdeA uses its mART domain to bind NAD+ and MARylate Ub on Arg42. The ADPr pyrophosphate bond is then cleaved by the PDE domain, and the resulting phosphoribosyl-Ub intermediate is transferred onto a substrate serine (Bhogaraju et al, 2016). *Chromobacterium violaceum* also secretes an effector protein, CteC, that MARylates Ub on Thr66 (Yan et al, 2020). In both cases, modification of Ub prevents activation and conjugation by E1 and E2 enzymes, respectively, resulting in significant impairments to host Ub signaling and many Ub-dependent cellular processes.

Eukaryotes also demonstrate an intricate relationship between ADP-ribosylation and ubiquitylation. A notable example is RNF146, an E3 ligase belonging to the really interesting new gene

[1]Department of Chemical Physiology and Biochemistry, Oregon Health & Science University, Portland, OR 97239, USA. [2]Department of Molecular Microbiology and Immunology, Oregon Health & Science University, Portland, OR 97239, USA. [3]Knight Cancer Institute, Oregon Health & Science University, Portland, OR 97239, USA.
✉E-mail: pruneda@ohsu.edu; cohenmic@ohsu.edu

(RING)-type class of E3 ligases, which catalyzes ubiquitylation of substrates in a PARylation-dependent manner, ultimately leading to their degradation by the Ub proteasome system (Zhang et al, 2011; Kang et al, 2011). Upon binding of iso-ADPr, an internal repeating unit of poly-ADP-ribose (PAR), to the Trp-Trp-Glu (WWE) domain of RNF146, an allosteric switch occurs that activates RNF146 ligase activity (DaRosa et al, 2015). A similar finding has been observed with E3 ligases containing PAR-binding zinc finger (PBZ) motifs, namely CHFR (Ahel et al, 2008; Kashima et al, 2012) and RNF114 (Li et al, 2023), which have been shown to ubiquitylate PARylated substrates (including PARP1) for degradation. Finally, DELTEX (DTX) E3 ligases highlight an even greater interplay between ubiquitylation and ADP-ribosylation. DTX family members (DTX1, 2, 3, 3L, 4) contain a DTX C-terminal (DTC) domain that has been shown to bind PAR and mediate PAR-dependent ubiquitylation of substrates (Ahmed et al, 2020). Conversely, the carboxy-terminus of Ub is MARylated upon heterodimerization of DTX3L and PARP9 (Yang et al, 2017). It was later shown that this Ub modification occurred with DTX3L alone, independent of PARP9 (Chatrin et al, 2020), making the role of ADP-ribosylation unclear in this model. Recently, Zhu et al proposed an intriguing model whereby DTX E3 ligases ubiquitylate the 3'-hydroxyl of the adenine-proximal ribose (A-ribose) of substrates, including free NAD$^+$ or ADPr (Zhu et al, 2022), and even ADP-ribosylated nucleic acids (Zhu et al, 2023). In this model, ADPr is ubiquitylated rather than Ub being ADP-ribosylated.

Using in vitro systems, these studies suggest an exciting and novel way for ADPr and Ub to occur at the same site on proteins as a dual PTM. However, a crucial question remained whether this dual PTM occurs on proteins within cells, and if it does, what significance it holds in terms of function. Herein, we show for the first time that a hydroxyl group within the A-ribose of MARylated PARP10 is polyubiquitylated in cells, generating a noncanonical ester-linked mono-ADPr Ub species (MARUbe). Moreover, we demonstrate that on PARP10 and PARP7, this dual PTM, which we call mono-ADP-ribosyl ubiquitylation (MARUbylation), is extended with K11-linked polyUb chains. Finally, we discovered that endogenous MARUbylation occurs in interferon-stimulated cells.

## Results

### PARP10 high molecular weight MARylation is dependent on ubiquitin

In 2008, PARP10 (formerly known as ARTD10 (Lüscher et al, 2021)) was the first member of the PARP family shown to possess MARylating activity (Kleine et al, 2008). Although when PARP10 was first discovered three years prior, it was originally proposed to possess auto-PARylating activity (Yu et al, 2005), as evidenced by weak detection with a monoclonal antibody specific for poly-ADPr (Yu et al, 2005; Kawamitsu et al, 1984). These experiments were done in vitro, so it was still unclear what kind of activity PARP10 displayed in cells, especially since MARylation had not been detected in cells at that time. This was largely due to the lack of antibodies/detection reagents specifically recognizing mono-ADPr versus poly-ADPr. Since then, new reagents specific for mono-

ADPr (Gibson et al, 2017; Bonfiglio et al, 2020; Longarini et al, 2023) and mono/poly-ADPr (Gibson et al, 2017; Lu et al, 2019) show that PARP10 is auto-MARylated in cells (Weixler et al, 2023). In fact, most of the PARP family members display MARylation activity in cells (Weixler et al, 2023). It was therefore striking to us that following expression of PARP10 using a GFP-PARP10 doxycycline (dox)-inducible HEK 293 cell line, we observed an upward mobility shift, or "smear", of ADP-ribosylation originating at the molecular weight (MW) of GFP-PARP10, detected using a mono/poly-ADPr antibody (Fig. 1A). This led us to conclude that, in our system, the high MW PARP10-mediated ADP-ribosylation smear is evidence of auto-PARylated PARP10. However, the high MW smear of ADP-ribosylation was also detected with a mono-ADPr-specific antibody (Bonfiglio et al, 2020), but not with a poly-ADPr-specific detection reagent (Bonfiglio et al, 2020; Gibson et al, 2017), indicating the absence of any PARylation (Fig. 1A). To confirm these antibodies/detection reagents were working as expected, we transiently overexpressed GFP-PARP10 in HEK 293 control and PARP1 KO cells to generate a MARylation signal or treated with a poly(ADP-ribose) glycohydrolase (PARG) inhibitor (James et al, 2016) to stabilize an endogenous PARP1-mediated PARylation signal (Fig. EV1). As expected, in the presence of PARG inhibitor (PARGi), we detected PARP1-dependent PARylation with the poly- and mono/poly-ADPr antibodies, but not with the mono-ADPr-specific antibody. Conversely, GFP-PARP10 MARylation was detected with the mono- and mono/poly-ADPr antibodies, but not with the poly-ADPr-specific antibody (Fig. EV1).

What could be underlying the high MW PARP10 MARylation smear, given that PARylation is not involved? PARP10 is the only PARP that contains Ub-interacting motifs (UIMs) (Suskiewicz et al, 2023), which have been shown in vitro to bind K63-linked tetraUb (Verheugd et al, 2013) and K48-linked diUb (Zhang et al, 2017), independently. PARP10 has also been shown to be polyubiquitylated by RNF114 in cells (Zhao et al, 2021). Therefore, we hypothesized that the high MW PARP10 MARylation smear could be due to polyubiquitylation. To test this idea, we incubated lysates containing dox-induced GFP-PARP10 with USP21, a nonspecific deubiquitinase (DUB) that removes all polyUb linkage types (Ye et al, 2011). Upon USP21 treatment, we observed a collapse of the high MW MARylated smear into a distinct mono-ADPr band at the MW of GFP-PARP10 (Fig. 1B), indicating that polyubiquitylation is responsible for the high MW PARP10 MARylation smear. We also wanted to determine which amino acid(s) on PARP10 are MARylated. Previous studies (Kleine et al, 2008; Vyas et al, 2014; Rosenthal et al, 2015; García-Saura and Schüler, 2021) demonstrated that acidic amino acids, such as glutamate (Glu) and aspartate (Asp), are MARylated on PARP10. Therefore, we treated lysates with hydroxylamine (buffered at pH 7.5), which rapidly cleaves ester-linked ADPr (Bredehorst et al, 1978; Riquelme et al, 1979) and observed removal of the MARylation smear (Fig. 1C). As a negative control, we subjected lysates to 300 mM HCl (pH 1.5) and observed no removal of MARylation, indicating that Glu/Asp-linked ADPr is stable in acidic conditions, as previously reported (Cervantes-Laurean et al, 1997). These results support the hypothesis that the high MW PARP10 MARylation smear is due to polyubiquitylation of PARP10 and that PARP10 MARylation occurs exclusively on Glu/Asp side chains in cells.

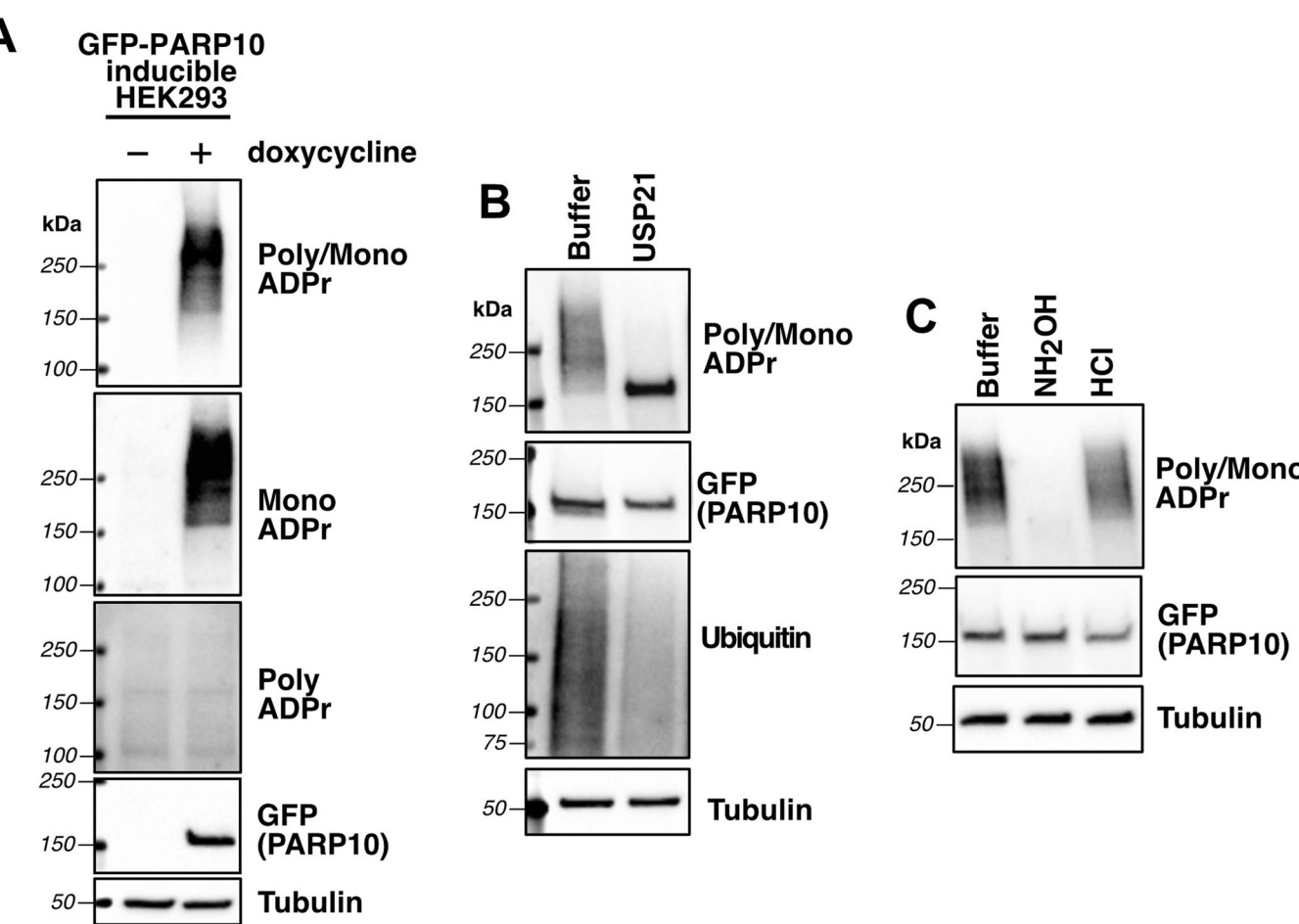

**Figure 1. High molecular weight PARP10 MARylation is ubiquitin dependent.**

(A) HEK 293 cells with doxycycline-inducible GFP-PARP10 were treated with doxycycline (10 µg/ml) for 24 h, followed by western blotting and probing for poly/mono ADPr (Cell Signaling Technology: E6F6A), mono-ADPr (Bio-Rad: HCA354), or poly-ADPr (Millipore Sigma: MABE1031). Representative western blot from *n* = 2 biological replicates. (B) HEK 293 cell lysates containing doxycycline-induced GFP-PARP10 were treated with 1 µM USP21 for 1 h at 37 °C, followed by western botting. Representative western blot from *n* = 2 biological replicates. (C) HEK 293 cell lysates containing doxycycline-induced GFP-PARP10 were treated with hydroxylamine (1 M, pH 7) or HCl (300 mM, pH 1.5) for 15 min, followed by methanol precipitation and western blotting (HCl-treated samples were neutralized with equimolar NaOH before precipitation). Representative western blot from *n* = 2 biological replicates. Source data are available online for this figure.

## An engineered bacterial DUB with profound Ub esterase activity cleaves a Ub ester-linked to ADPr on a model substrate

There are two major ways proteins can be modified with Ub. One is through a lysine, which occurs via a canonical isopeptide linkage. The other is ubiquitylation of serine or threonine, generating an ester-linkage (or with a cysteine, leading to a thioester). Ester-linked Ub may also be conjugated to proteins through ubiquitylation of the covalently attached ADPr group, as previously suggested in vitro (Zhu et al, 2022). While hydroxylamine is often used to diagnostically release ester-linked ubiquitylation (Rehman et al, 2024; McCrory et al, 2022; Cesare et al, 2021), in our case, its use would be confounded by our finding that PARP10 is Glu/Asp-MARylated in cells, a linkage that is also labile to hydroxylamine treatment. Recently, the bacterial DUB TssM from *Burkholderia pseudomallei* was shown to reverse ester-linked lipopolysaccharide ubiquitylation as a strategy to evade host cell-intrinsic immune responses (Szczesna et al, 2024). Using Lys-Ub and Ser/Thr-

Ub models of isopeptide- and ester-linked Ub, respectively, TssM was 30 times more active towards ester-linked Ub than isopeptide-linked Ub. Guided by a Ub-bound crystal structure of TssM, a V466R variant (referred to here as TssM*) was identified that further uncoupled esterase and isopeptidase activities, enhancing its esterase-specific activity by ~5000-fold (Szczesna et al, 2024). We, therefore, reasoned that TssM and TssM* could potentially be used to determine if PARP10 is ubiquitylated on Glu/Asp-MARylation sites in cells.

Before testing TssM and TssM* on MARylated PARP10 isolated from our dox-inducible cells, we wanted to validate their specificities against ADPr and NAD$^+$ containing an ester-linked Ub at the 3'-OH of the A-ribose (Fig. 2A). Following Zhu et al (Zhu et al, 2022), we used DTX2 to ubiquitylate NAD$^+$ or ADPr (Fig. 2A). In addition to ester-linked ubiquitylation of NAD$^+$, DTX2 was also reported to catalyze isopeptide-linked auto-ubiquitylation in the same reaction (Zhu et al, 2022). We also detected this activity, a convenient internal control for DUB specificity (Fig. 2B). We found that DTX2 could readily conjugate

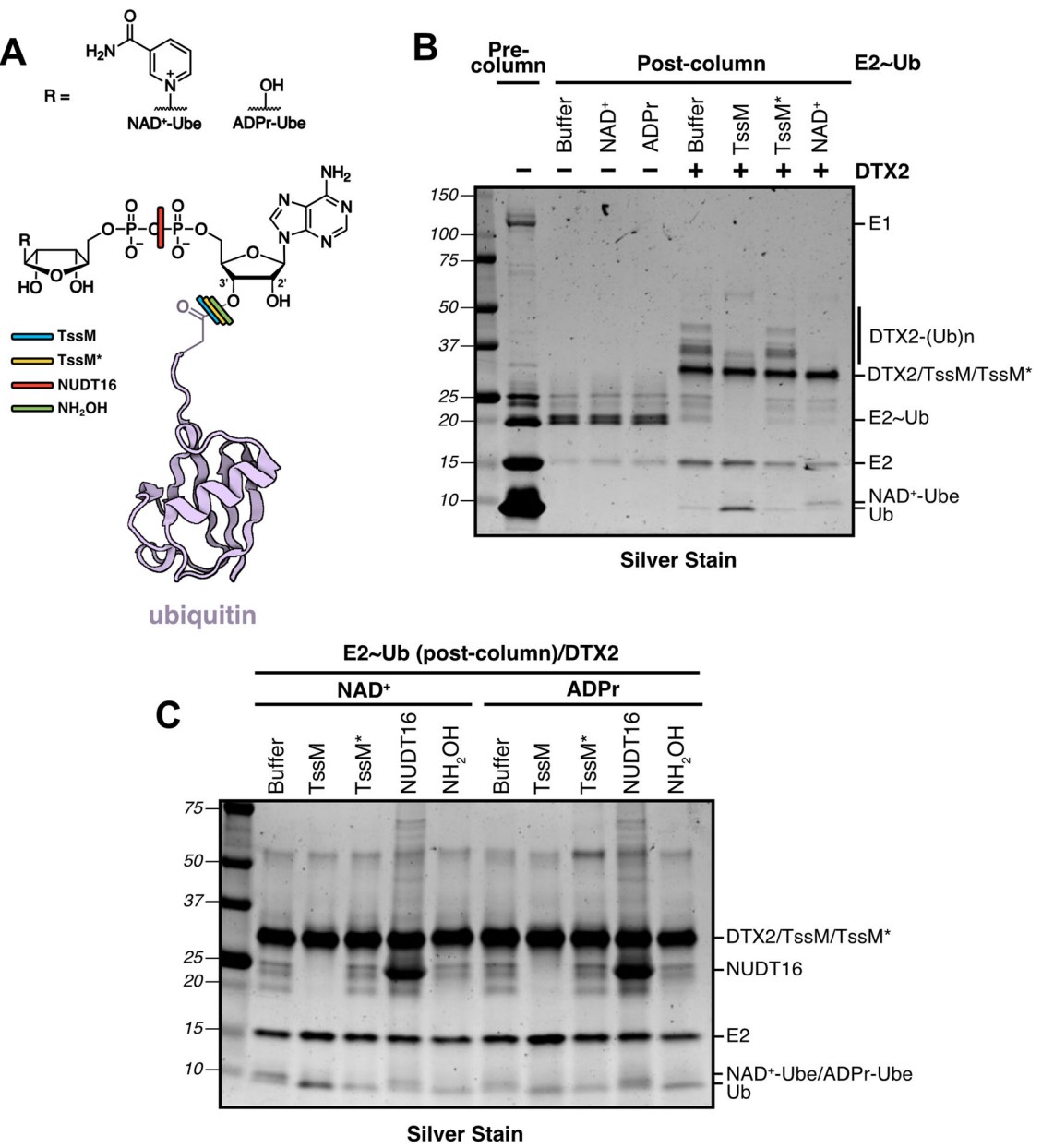

**Figure 2. TssM\* is highly specific for removal of ester-linked Ub on NAD+ and ADPr.**

(A) Structure of Ub ester-linked to the A-ribose 3'-OH of NAD+ (NAD+-Ube) and ADPr (ADPr-Ube), along with corresponding cut sites for different enzymatic or chemical treatments. (B) E2-Ub conjugate was generated and purified by size exclusion chromatography, followed by adding substrates (NAD+ or ADPr) or DUBs (TssM or TssM\*) in the presence or absence of DTX2. (C) Purified E2-Ub was incubated with DTX2 and substrates (NAD+ or ADPr), followed by enzymatic (TssM, TssM\*, NUDT16) or hydroxylamine (1 M pH 7.4) treatment. Representative gels shown are from n = 2 biological replicates. Source data are available online for this figure.

Ub to NAD+ and ADPr to form Ub ester-linked NAD+ (NAD+-Ube) and ADPr (ADPr-Ube), as demonstrated by the upward shift of the Ub band visualized by silver staining (Fig. 2B,C). As expected, we found that TssM, but not TssM\*, cleaved DTX2 auto-ubiquitylation (Fig. 2B), suggesting DTX2 auto-ubiquitylation is isopeptide-linked and that TssM\* is indeed a valuable tool for ester-specific Ub removal. TssM and TssM\* efficiently cleaved the Ub ester of NAD+-Ube and ADPr-Ube (Fig. 2C). Similar to TssM and TssM\*, hydroxylamine eliminated the upper bands corresponding to NAD+-Ube and ADPr-Ube, confirming Ub is attached

to NAD+ and ADPr via an ester linkage (Fig. 2C). Finally, we treated NAD+-Ube and ADPr-Ube with Nudix-Type Motif 16 (NUDT16), a hydrolase that cleaves MARylation and PARylation at the pyrophosphate bond in ADPr (Palazzo et al, 2015). NUDT16 failed to shift the NAD+-Ube and ADPr-Ube bands to the faster-migrating Ub form (Fig. 2C), demonstrating that the Ub is ester-linked to the NUDT16-cleaved NAD+ and ADPr, as shown previously (Zhu et al, 2022). However, we cannot exclude the possibility of Ub attachment occurring on a hydroxyl group of the distal ribose of NAD+ or ADPr.

To further validate the esterase-specific activity of TssM*, we used maltoheptaose-Ub, another sugar-containing ester-linked Ub substrate. Following Kelsall et al (Kelsall et al, 2022), we generated maltoheptaose-Ub using the E3 ligase HOIL-1 in the presence of K63-linked diUb, resulting in ubiquitylation of the C6 hydroxyl group of a glucosyl unit in maltoheptaose (Fig. EV2A). Similar to NAD$^+$-Ube and ADPr-Ube, the maltoheptaose-Ub signal was removed by TssM, TssM*, and hydroxylamine. The isopeptide-linked K63 diUb substrate (serving as an internal specificity control), however, was only cleaved by TssM (Fig. EV2B). Taken together, these results confirm that TssM is an active DUB with dual esterase and isopeptidase activity and that the TssM* variant displays exquisite esterase-specific DUB activity for Ub attached to hydroxyl groups on different types of sugars.

## Enzymatic and chemical sensitivity studies reveal that PARP10 is ubiquitylated on MARylated Glu/Asp sites in cells

Having validated TssM* as an esterase-specific DUB capable of cleaving Ub ester-linked to ADPr in vitro, we asked if PARP10 is ubiquitylated at the A-ribose 3'-OH group of Glu/Asp-MARylation sites in cells. We transiently expressed tagged Ub (HA-Ub) in our GFP-PARP10 dox-inducible cells and immunoprecipitated GFP-PARP10 using GFP-trap beads. After stringent washes (with 7 M urea and 1% SDS) to remove noncovalently interacting ubiquitylated proteins, we performed on-bead enzymatic or chemical treatments (Fig. 3A), followed by western blotting to detect changes in MARylation and ubiquitylation on PARP10 (Fig. 3B). Treatment

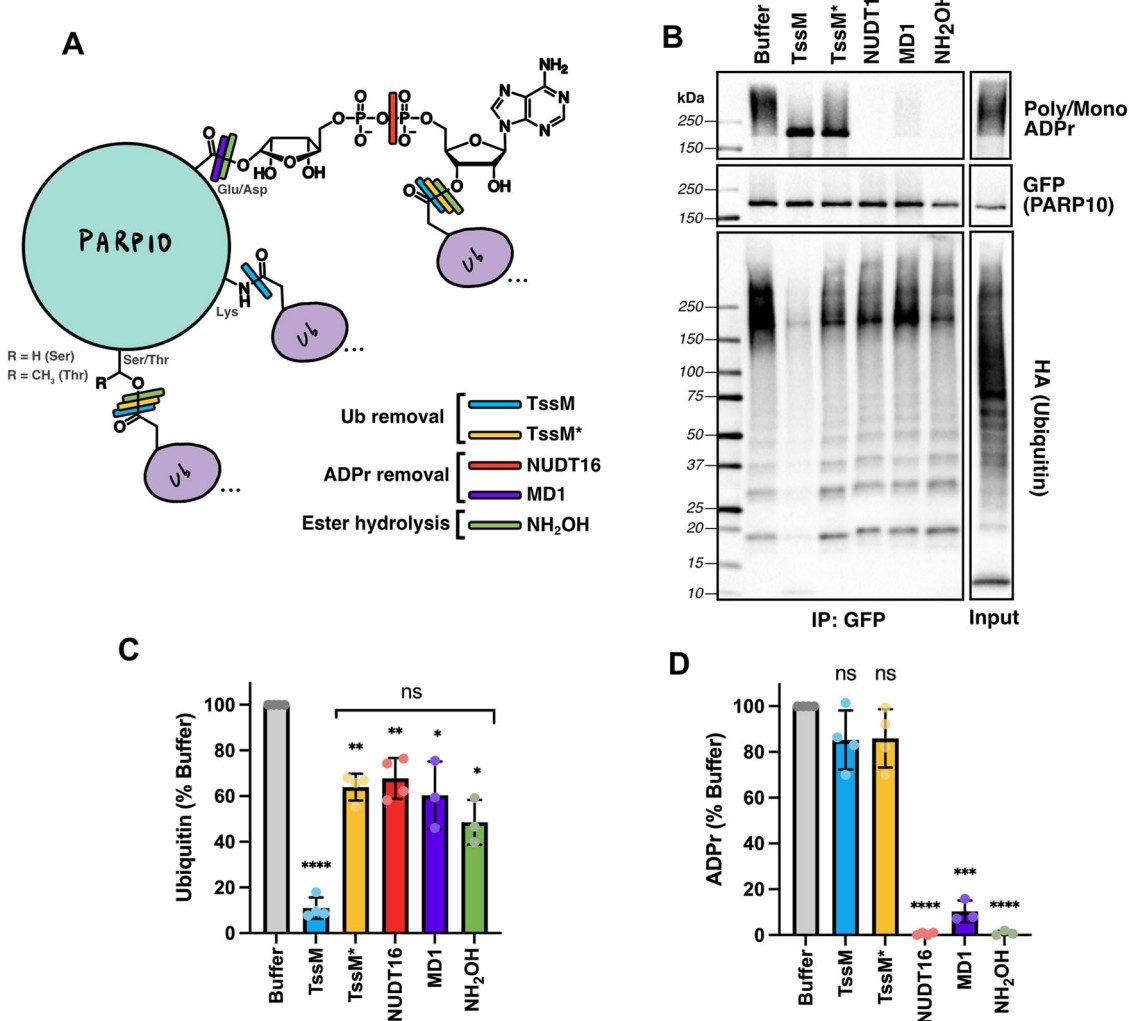

**Figure 3. Polyubiquitin is directly attached to a hydroxyl group on ADP-ribose of MARylated PARP10 in cells.**

(A) Schematic of potential sites for Ub attachment on MARylated PARP10, with corresponding cut sites for different enzymatic or chemical treatments. (B) GFP-PARP10, from doxycycline-treated HEK 293 cells transfected with HA-Ub, was immunoprecipitated with GFP-trap beads, washed stringently (7 M Urea, 1% SDS), and beads were treated with DUBs, ADPr hydrolases, or chemicals, followed by western blotting. Quantification of the ~150 kDa ubiquitin smear (C) and ADPr smear (D) from (B); $n = 3$–4 biological replicates. Asterisks indicate statistical significance (ns: not significant). For (C), TssM: ****$P < 0.0001$, TssM*: **$P = 0.0012$, NUDT16: **$P = 0.0055$, MD1: *$P = 0.0431$, NH$_2$OH: *$P = 0.0119$, calculated using a one-sample $t$ and Wilcoxon test relative to buffer treatment. Comparison between TssM*, NUDT16, MD1, and NH$_2$OH was measured by one-way ANOVA (ns). For (D), NUDT16: ****$P < 0.0001$, MD1: ***$P = 0.0009$, NH$_2$OH: ****$P < 0.0001$, calculated using a one-sample $t$ and Wilcoxon test relative to buffer treatment. Source data are available online for this figure.

with TssM removed nearly all Ub from PARP10, resulting in a collapse of the high MW ADPr smear to a single band at the MW of GFP-PARP10, as seen with USP21 treatment (Fig. 1C). Remarkably, treatment with TssM* removed ~40% of the Ub from PARP10, partially collapsing the high MW ADPr smear into a single band (Fig. 3B,C). These results show that a significant population of Ub on PARP10 is ester-linked in cells.

Ester-linked Ub could be on Ser/Thr residues or the 3'-OH of the attached ADPr (Fig. 3A). To distinguish these possibilities, we treated samples with NUDT16 (Fig. 3A). If NUDT16 decreases the high MW PARP10 Ub smear, this would support the hypothesis that Ub is ester-linked to the A-ribose 3'-OH of ADPr. If no change in PARP10 ubiquitylation is observed with NUDT16 treatment, this would suggest that Ub is ester-linked to a Ser/Thr on PARP10. Upon treatment with NUDT16, we observed complete removal of MARylation (Fig. 3B,D) and a ~40% reduction in Ub, similar to TssM* treatment (Fig. 3B,C), suggesting that Ub is indeed linked to the A-ribose 3'-OH of ADPr. Furthermore, we found that treatment with hydroxylamine or MacroD1 (MD1), an ADPr hydrolase that cleaves Glu/Asp-linked ADPr (Jankevicius et al, 2013; Rosenthal et al, 2013), also resulted in a ~40% reduction in Ub (Fig. 3C), with near complete removal of MARylation (Fig. 3D). We also tested other macrodomain-containing ADPr hydrolases (MacroD2 (MD2) and terminal ADP-Ribose protein glycohydrolase (TARG) (Sharifi et al, 2013)), and observed a similar effect, with TARG displaying more potent removal of PARP10 MARylation and subsequent Ub release (Fig. EV3). Together, these results demonstrate that PARP10 is polyubiquitylated in cells, and nearly half of this polyUb is ester-linked to the A-ribose 3'-OH of Glu/Asp-MARylated PARP10. We refer to this dual modification as a mono-ADPr-Ub ester (MARUbe) and the novel PTM as mono-ADP-ribosyl ubiquitylation, or MARUbylation.

## MARUbylation contains K11-linked polyubiquitin

We next sought to further characterize the population of ester-linked polyUb on ADPr and the ~60% polyUb remaining on PARP10 after TssM* or ADPr hydrolase treatment. To distinguish these two populations, we stringently washed immunoprecipitated GFP-PARP10 as described above; however, after on-bead treatment with NUDT16, we separated the supernatant (containing polyUb released from ADPr) and beads (containing polyUb retained on PARP10) (Fig. 4A). These fractions were then treated with the nonspecific DUBs, TssM and USP21, or a panel of polyUb linkage-specific DUBs in a workflow termed Ubiquitin Chain Restriction Analysis (UbiCRest) (Mevissen et al, 2013; Hospenthal et al, 2015) (Fig. 4B). In the beads fraction, the remaining polyUb on PARP10 after NUDT16 treatment was efficiently removed by the nonspecific DUBs, but not with TssM*, suggesting canonical isopeptide-linked polyUb. Intriguingly, this polyUb species was not removed by any linkage-specific DUBs, suggesting it is either highly complex polyubiquitylation or multi-mono-ubiquitylation. As an alternative strategy to characterize the isopeptide-linked polyUb on PARP10 and to test if it is dependent on MARylation, we inhibited MARylation in cells before immunoprecipitation using a pan-MARylating PARP inhibitor, RBN010860 (Wigle et al, 2020). RBN010860 potently inhibited PARP10 MARylation with a half-maximal inhibitory concentration ($IC_{50}$) of 51.4 nM (Fig. EV4A–C), similar to the reported $K_d$ of 34 nM measured for PARP10 in vitro

(Wigle et al, 2020). We observed that even after near complete inhibition of PARP10 MARylation using RBN010860, a polyUb signal on PARP10 was still present and was cleaved by TssM but not TssM*, suggesting an isopeptide-linked polyUb species that is not dependent on MARylation (Fig. EV4D,E).

In the supernatant fraction, we observed a release of polyUb upon NUDT16 treatment, with the major species being a diUb linkage. From the polyUb linkage-specific DUBs tested, only Cezanne could significantly cleave this diUb species (Fig. 4A). Cezanne/OTUD7B, a member of the ovarian tumor (OTU) family of DUBs, was the first DUB shown to cleave K11-linked polyUb preferentially (Bremm et al, 2010; Mevissen et al, 2013; Hospenthal et al, 2015; Mevissen et al, 2016; Bonacci et al, 2018). We confirmed this UbiCRest cleavage profile using a recombinant K11-linked diUb substrate, demonstrating that only Cezanne and the nonspecific TssM or USP21 could cleave K11-linked polyUb under our conditions (Fig. 4C). Taken together, these results indicate that PARP10 contains two polyUb species: (i) K11-linked polyUb attached to MARUbe and (ii) canonical isopeptide-linked polyUb with an unknown, likely complex, chain identity.

We wondered if K11-linked MARUbylation was unique to PARP10 or if it occurs in other PARP family members. PARP7 has been shown to interact with many E3 ligases (DTX2, HUWE1, RNF114) and promote MARylation-dependent ubiquitylation of substrates, including itself, leading to proteasomal degradation (Zhang et al, 2020). To test if PARP7 undergoes K11-linked MARUbylation, we transiently co-expressed GFP-PARP7 and HA-Ub in HEK 293 T cells and immunoprecipitated GFP-PARP7, followed by NUDT16 treatment to release polyUb attached to ADPr. Like PARP10, NUDT16 treatment released a major diUb species that was cleaved by Cezanne, demonstrating that PARP7 is MARUbylated and contains K11-linked polyUb (Fig. EV4F). Together, these results suggest that K11-linked MARUbylation might be a common dual modification of PARPs.

## Type I interferon drives endogenous MARUbylation

While the experiments above demonstrated that exogenously expressed PARPs can undergo MARUbylation, we sought to determine whether this modification also occurs on endogenous PARPs. Most MARylating PARPs are absent or are lowly expressed under basal conditions (Cho et al, 2022; Sanderson and Cohen, 2020). Major inducers of MARylating PARP expression are interferons (IFNs) in various contexts, from viral infection to cancer-immune cell signaling (Biaesch et al, 2023; Brooks et al, 2023). For example, PARP10 and PARP14 are highly upregulated by the type I interferon, IFN-β, in pancreatic cancers (Moore et al, 2021). Additionally, it has recently been shown that PARP14 expression and catalytic activity (MARylation) is strongly induced upon IFN-γ treatment (Kar et al, 2024; Ribeiro et al, 2024). Therefore, we wondered whether IFN-β treatment would induce PARP10 expression and allow for the detection of endogenous MARUbylation. For our experiments, we used PARP1 knockout HEK 293 cells to eliminate PARP1-mediated ADP-ribosylation, as the basal activity of PARP1 could hinder our ability to detect IFN-β-induced endogenous MARylation. Consistent with prior work (Moore et al, 2021), we found that the treatment of PARP1 KO HEK 293 cells with IFN-β strongly induced the expression of PARP10 and PARP14 (Fig. 5A). The levels of GFP-PARP10 in our

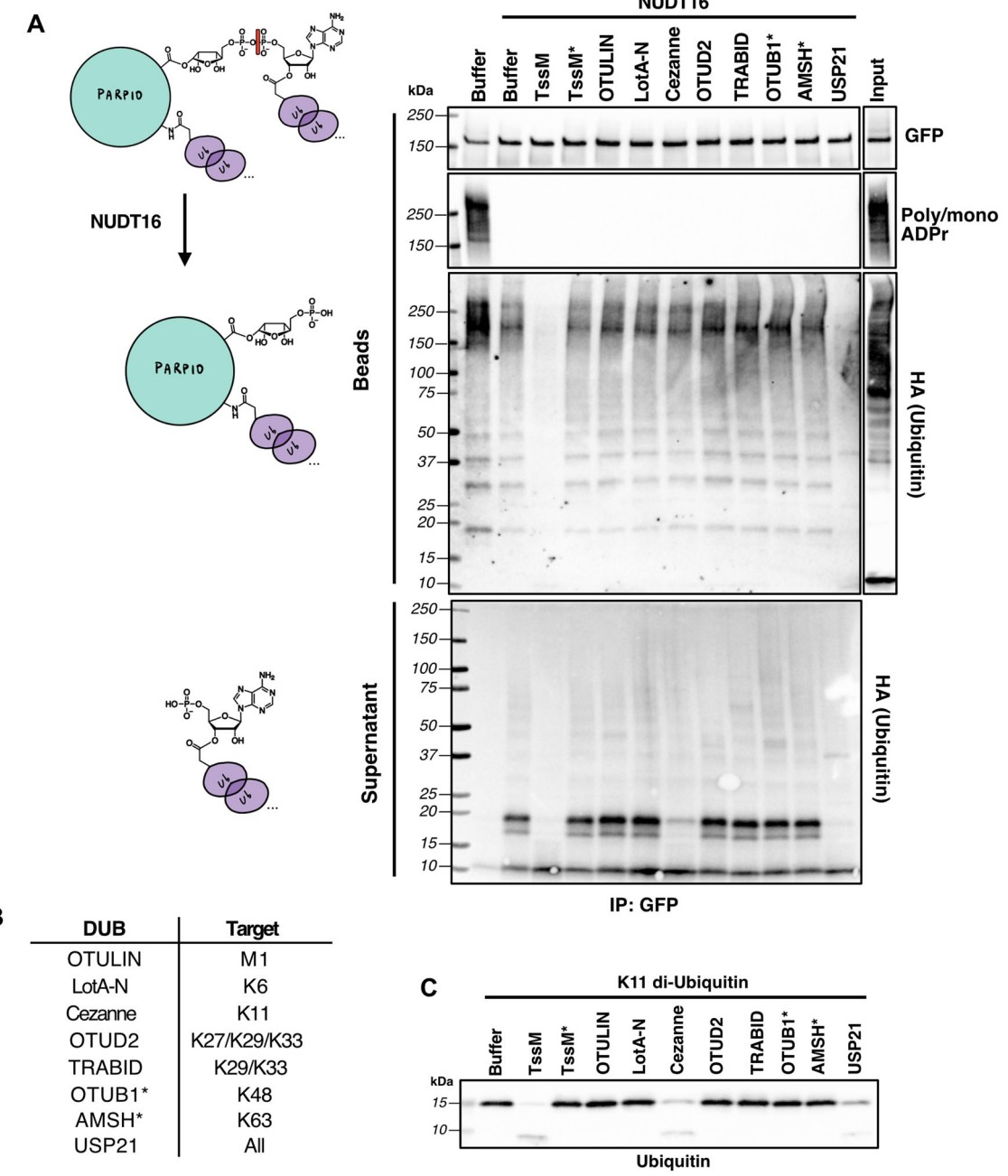

**Figure 4. PARP10 MARUbe contains K11-linked Ub chains.**

(A) GFP-PARP10, from doxycycline-treated HEK 293 cells transfected with HA-Ub, was immunoprecipitated with GFP-trap beads, washed stringently (7 M Urea, 1% SDS), and beads were treated with NUDT16. The supernatant and beads were separated and further treated with polyubiquitin linkage-specific DUBs, followed by western blotting. Representative western blot from n = 2 biological replicates. (B) Table of polyubiquitin linkage-specific DUBs and their preferential cut site. (C) Positive control experiment with polyubiquitin linkage-specific DUBs using K11 diUb substrate (1 µM). Representative western blot from n = 1 biological replicate. Source data are available online for this figure.

doxycycline-inducible cell line were comparable to those of endogenous PARP10 induced by IFN-β (Fig. EV5A). In addition to increasing the protein levels of PARP10 and PARP14, IFN-β strongly induced MARylation that appeared as a smear at a high MW (Fig. 5A).

We next asked if the IFN-β-induced ADP-ribosylation was due to the activity of MARylating PARPs, particularly PARP10 and PARP14 given their strong induction by IFN-β. Treatment of cells with 1 µM RBN010860, which completely inhibits PARP10 activity in cells (Fig. EV4), abolished IFN-β-induced ADP-ribosylation (Fig. 5B). On the other hand, treatment of cells with the PARP14-specific inhibitor, 0.1 µM RBN012759, substantially diminished the high MW IFN-β-induced MARylation smear, but revealed a robust MARylation smear that originated ~140 kDa (Fig. 5B). The

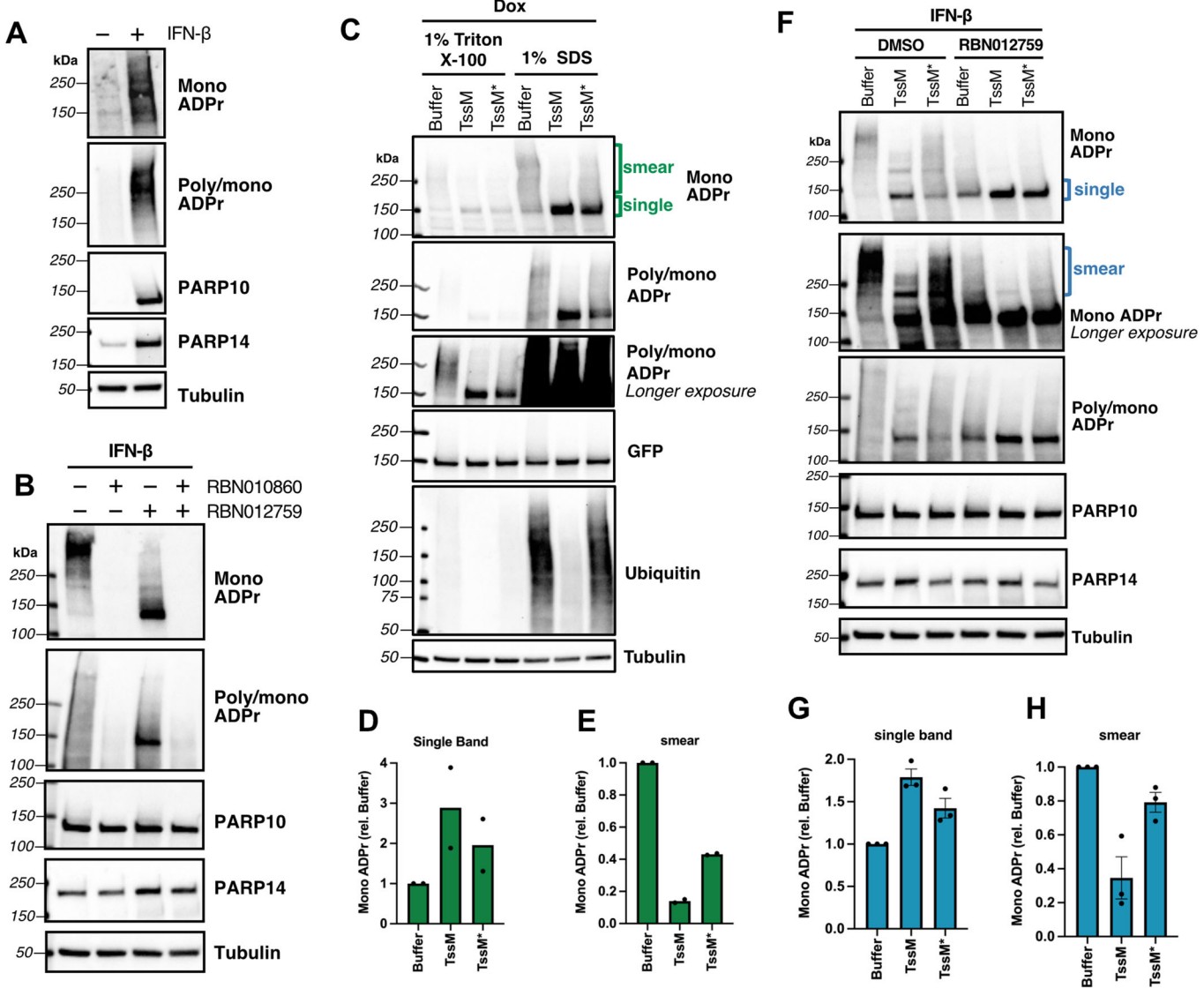

**Figure 5. IFN-β induces endogenous MARUbylation of PARP10 and PARP14.**

(A) PARP1 KO HEK 293 cells were treated with IFN-β (140 U/ml) for 24 h and lysed in CLB (contains 1% TX-100) followed by western blotting (no boiling of samples). Representative western blot from $n = 3$ biological replicates. (B) PARP1 KO HEK 293 cells were co-treated with IFN-β (140 U/ml) and RBN010860 (1 μM) or RBN012759 (0.1 μM) for 24 h and lysed in CLB with 1% SDS followed by western blotting (no boiling of samples). Representative western blot from $n = 2$ biological replicates. (C) HEK 293 GFP-PARP10 doxycycline-inducible cells were treated with doxycycline (10 μg/ml) for 24 h, followed by lysis in CLB with 1% Triton X-100 or 1% SDS. Lysates were diluted 6-fold and TssM DUBs (1 μM) were added for 1 h at 37 °C followed by western blotting (no boiling of samples). Quantification of the mono ADPr ~150 kDa single band (D) and smear (E) from (C); $n = 2$ biological replicates. (F) PARP1 KO HEK 293 cells were treated with IFN-β (140 U/ml) $+/-$ RBN012759 (0.1 μM) for 24 hr, followed by lysis in CLB with 1% SDS. Lysates were diluted 6-fold and TssM DUBs (1 μM) were added for 1 h at 37 °C followed by western blotting (no boiling of samples). Quantification of the mono ADPr ~150 kDa single band (G) and smear (H) from (F); $n = 3$ biological replicates. Source data are available online for this figure.

appearance of the ~140 kDa MARylation signal following RBN012759 treatment could be due to increased NAD$^+$ availability for another MARylating PARP, like PARP10, as a consequence of preventing NAD$^+$ consumption by PARP14. Indeed, a previous study demonstrated that PARP14 significantly contributes to IFN-β-mediated NAD$^+$ consumption (Moore et al, 2021). The combination of RBN010860 and RBN012759 abolished the ~140 kDa IFN-β-induced MARylation signal that was observed with RBN012759 treatment alone (Fig. 5B). Taken together, these results show that IFN-β-induced MARylation in HEK 293 cells is

mediated by PARP14 and another MARylating PARP. Since the ~140 kDa IFN-β-induced MARylation signal corresponds to the position of endogenous PARP10 on an SDS-PAGE gel, we hypothesize that this MARylation signal represents endogenous auto-MARylated PARP10.

We next determined if treatment with the DUBs TssM and TssM* would collapse the IFN-β-induced endogenous MARylation smear. The collapse of this MARylation smear would indicate that IFN-β can induce endogenous MARUbylation. Initially, we wanted to examine the activity of TssM and TssM* in two different lysis

conditions: CLB (cell lysis buffer containing 1% TX-100) or CLB with 1% SDS. For these experiments, we used the GFP-PARP10 dox-inducible cells. Incorporating 1% SDS into CLB enhanced detection of the MARylation smear compared to CLB alone (Fig. 5C). Lysates from both conditions were diluted 6-fold before treatment with DUBs. TssM, and, to a lesser extent, TssM* collapsed the MARylation smear and simultaneously enhanced the prominence of a single band at the molecular weight of GFP-PARP10 (Fig. 5C–E), consistent with previous results. We then examined their effects on the IFN-β-induced MARylation smear in lysates (lysed in 1% SDS CLB) derived from cells treated with DMSO control or RBN012759. In DMSO control-treated cells, TssM and TssM*, though to a lesser degree, diminished the high MW IFN-β-induced MARylation smear while enhancing the appearance of two single MARylation bands ~230 kDa and ~140 kDa, which correspond to the positions of endogenous PARP14 and PARP10, respectively, on an SDS-PAGE gel (Fig. 5F). Similar effects were observed in RBN012759-treated cells. However, the ~140 kDa was more prominent, and the ~230 kDa was nearly undetectable (Fig. 5F–H). These results strongly indicate that IFN-β can drive endogenous MARUbylation, likely occurring on PARP14 and PAPR10; however, we cannot exclude the possibility that IFN-β-induced MARUbylation occurs on other proteins, perhaps targets of PARP14 and PARP10.

### Inhibition of the ubiquitin-proteasome pathway alters the cellular levels of MARUbylation

Finally, we explored the functional connection between the ubiquitin-proteasome pathway and MARUbylation. In GFP-PARP10 HEK 293 cells, we found that inhibition of the proteasome with either MG132 (Tsubuki et al, 1996) or bortezomib (Hideshima et al, 2001) increased the levels of the MARylation smear in a time-dependent manner (Fig. EV5B). Although proteasomal inhibition did not significantly increase the levels of GFP-PARP10, we did observe an increase in a GFP-positive smear just above the molecular weight of GFP-PARP10, consistent with the accumulation of polyUb conjugates on GFP-PARP10 (Fig. EV5B). Treatment of GFP-PARP10 cells with the ubiquitin activation inhibitor TAK-243 (Hyer et al, 2018) resulted in the time-dependent collapse of the MARylation smear and an increase in a single band at the molecular weight of GFP-PARP10 (Fig. EV5B). These results provide further evidence that the MARylated form of GFP-PARP10 is enriched with ubiquitin, consistent with MARUbylation. The underlying cause of the increased MARUbylation upon proteasomal inhibition is unclear. One possibility is that proteasome inhibition stabilizes a repressor of MARUbylation removal (e.g., an ADPr hydrolase or DUB).

We next examined the effects of ubiquitin-proteasome inhibition on IFN-β-induced endogenous MARylation in PARP1 KO HEK 293 cells. TAK-243 treatment eliminated the MARylation smear and triggered the emergence of single MARylated bands at the molecular weights of PARP14 and PARP10 (Fig. EV5C). In cells also treated with the PARP14-specific inhibitor RBN012759, the PARP14 MARylated band significantly reduced, whereas the PARP10 MARylated band increased (Fig. EV5C). TAK-243 treatment also decreased the smear just above PARP10 (Fig. EV5C). Treatment with bortezomib did not significantly increase the pattern of MARylation but did increase the smear just above

PARP10 (Fig. EV5C). These results further support that endogenous MARUbylation of PARP10 and PARP14 can occur in IFN-β-stimulated cells.

## Discussion

The prevailing view in biology is that a PTM-targeted amino acid is modified with a single PTM (e.g., methylation of lysine, phosphorylation of serine) or, in some cases, multiple PTMs of the same type (e.g., di- and trimethylation of lysines). Intriguingly, however, there are situations where a single residue could be dually modified with two distinct PTMs. One study showed that a single histone lysine can contain acetyl and methyl groups, forming a novel dual modification called acetyl-methyllysine (Kacme) (Lu-Culligan et al, 2023). Another dual-type modification is protein pyrophosphorylation, where a phosphorylated serine is phosphorylated again, generating pyrophosphoserine (Bhandari et al, 2007; Morgan et al, 2024). Lastly, Ub on ubiquitylated targets can subsequently undergo acetylation on lysine (Lys) or phosphorylation on serine, threonine, or tyrosine (Ser/Thr/Tyr) (Swatek and Komander, 2016). In this report, we show, for the first time, that exogenously expressed PARP10 is MARUbylated on Glu/Asp sites in cells. This dual modification, MARUbe, contains a Ub ester conjugated to the mono-ADPr attached to Glu/Asp sites on PARP10 and PARP7. We further reveal that endogenous MAR-Ubylation can occur in response to IFN-β, emphasizing the significance of this dual modification.

We observed a greater degree of MARUbylation when PARP10 was expressed using a dox-inducible system compared to transient overexpression, which instead led to more of a single MARylated band at the molecular weight of GFP-PARP10 (Figs. 1 and EV1). One explanation for this difference could be the higher levels of PARP10 obtained with transfection compared to the dox-inducible system, leading to supraphysiological MARylation on non-Glu/Asp sites. Another observation is that while PARP10 auto-MARylation is highly mass-shifted (detected by poly/mono-ADPr antibody), the overall levels of PARP10 are not (detected by GFP antibody), potentially suggesting only a small fraction of PARP10 molecules are modified in this manner (Fig. 3). Indeed, proteomic studies characterizing endogenous PARP1-mediated ADP-ribosylation revealed less than 11% of ADP-ribosylation occupancy in half of the ADP-ribosylated sites identified (Martello et al, 2016). In contrast, bacterial ADP-ribosyltransferases modify their substrates much more efficiently, with some showing a 1:1 stoichiometry for target residue modification (Ganesan et al, 1999). This highlights the more transient and dynamic nature of eukaryotic ADP-ribosylation compared to bacterial ADP-ribosylation, which may be governed by evolution for substrate specificity, amino acid preference, or susceptibility to ADPr hydrolases (Cohen and Chang, 2018). An alternative explanation for the low stoichiometry of PARP10 modification could be due to the inherent lability of esters during sample preparation. This is evident from recent reports showing that ester-linked PTMs, such as Glu/Asp-ADPr or ester-linked ADPr-Ub, are highly sensitive to heat and alkaline conditions (Tashiro et al, 2023; Longarini and Matić, 2024). Inspired by these studies, we found that adding 1% SDS to the lysis buffer, which presumably inactivates potential ADPr hydrolases and DUBs, and avoiding the boiling step before resolving proteins

on SDS-PAGE provided optimal conditions for detecting IFN-β-stimulated endogenous MARUbylation (Figs. 5 and EV5).

A key discovery in our study is that MARUbylation occurs endogenously in response to IFN-β (Figs. 5 and EV5). Our findings, using a pan-PARP inhibitor that inhibits PARP10 (RBN010860) and a PARP14-selective inhibitor (RBN012759), along with the significant upregulation of PARP10 and PARP14, suggest that PARP10 and PARP14 are critical regulators of IFN-β-stimulated endogenous MARUbylation. Moreover, the MW of the bands detected by western blot using ADPr antibodies supports the notion that PARP10 and PARP14 are primary targets of MARUbylation. Because RBN010860 inhibits other MARylating PARPs beyond PARP10, we can't exclude the possibility that endogenous MARUbylation is mediated by PARP14 and another MARylating PARP. The development of PARP10-selective inhibitors will aid in clarifying the specific role of PARP10 in cells, as well as a more general role for MARUbylation in interferon-primed immune responses.

What is the E3 ligase that attaches ubiquitin to MARylated proteins to generate MARUbe under IFN-β stimulation? Thus far, DTXs are the only E3 ligases shown to generate MARUbe in vitro (Zhu et al, 2022, 2023). Whether any DTX family members serve as the physiological E3 ligases responsible for MARUbylation under IFN-β stimulation remains to be determined. Beyond DTXs, other E3 ligases have been shown to ubiquitylate unconventional substrates. For example, RanBP-type and C3HC4-type zinc finger-containing protein 1 (RBCK1; also known as HOIL-1) was shown to ubiquitylate glycogen on a hydroxyl moiety to generate an ester-linked ubiquitin (Kelsall et al, 2022). The shared carbohydrate-like chemical properties of glycogen and ADP-ribose raises the possibility that HOIL-1 could ubiquitylate MARylated proteins. Understanding how MARUbylation is produced in the context of IFN-β stimulation, and possibly even removed through the action of specific DUBs, will be an important area of future work.

What is the physiological function of MARUbylation? Our finding that PARP10 and PARP7 MARUbylation contains K11-linked polyUb provides some clues to its physiological function. Although less well understood than the canonical K48- and K63-linked polyUb chains, several studies have suggested that K11-linked polyubiquitylation plays a role in signaling and protein degradation (Swatek and Komander, 2016). For example, K11-linked polyubiquitylation of receptor-interacting protein 1 (RIP1) upon tumor necrosis factor-α (TNF-α) stimulation regulates NF-κB activation (Dynek et al, 2010). In another study, K11-linked polyubiquitylation of mitotic proteins led to their degradation during mitosis (Matsumoto et al, 2010). Linking it more closely with ADP-ribosylation, a recent study revealed that the E3 ligase RNF166, along with its paralog RNF114, mediates K11-linked polyubiquitylation of PARP5a (also known as tankyrase 1) in a manner dependent on PARP5a activity (Perrard and Smith, 2023). This study further showed that the Di19 domain of RNF166 binds to MARylated PARP5a (Perrard and Smith, 2023); a separate independent study demonstrated that the Di19 domain of RNF114 binds to a MARylated peptide (Longarini et al, 2023). These studies suggest the intriguing possibility that the Di19 and UIM domains of RNF114 could act as a MARUbe reader, enabling RNF114/166 to extend the Ub chain to generate K11-linked MARUbylation.

Beyond K11-linked MARUbylation, PARP10 also contains isopeptide-linked Ub, whose identity could not be conclusively identified using the UbiCRest assay. It is possible that a more complex species of Ub, including branched, mixed, or multi-monoUb, could explain why we do not see removal of the isopeptide-linked Ub on PARP10 with any of the linkage-specific DUBs (Fig. 4A). To gain a complete understanding of PARP10's function in cells, it will be crucial to determine the molecular nature of isopeptide-linked Ub on PARP10 and the interplay between isopeptide-linked polyubiquitylation and MARUbylation.

This study adds to the growing crosstalk between ADP-ribosylation and ubiquitylation, highlighting the remarkable complexity of PTMs and their influence on signaling pathways. While there are still many unanswered questions, we can say that *all good things come in pairs*…of esters; one for ADP-ribosylation and one for ubiquitylation.

# Methods

### Reagents and tools table

| Reagent/resource | Reference or source | Identifier or catalog number |
|---|---|---|
| **Experimental models** | | |
| HEK 293T (*H. sapiens*) | ATCC | CRL-3216 |
| HEK 293 control and PARP1 KO (*H. sapiens*) | Shrestha et al, 2016 https://doi.org/10.1074/jbc.M116.726729 | Prof. Michael Garabedian, NYU Langone, USA |
| **Recombinant DNA** | | |
| pET28b-DTX2$^{389-622}$ (*H. sapiens*) | This study | N/A |
| pET28a-UBE2D3 (*H. sapiens*) | Gift from R. Klevit | N/A |
| pET17b-Ub (*H. sapiens*) | Gift from D. Komander | N/A |
| pET21d-UBE1 (*H. sapiens*) | | Addgene #34965A |
| pETM30-TssM and TssM* (*B. pseudomallei*) | Szczesna et al, 2024 | N/A |
| pOPINS-USP21$^{196-565}$ (*H. sapiens*) | Ye et al, 2011 | Addgene #61585 |
| pOPINE-Cezanne$^{129-438}$ (*H. sapiens*) | Mevissen et al, 2016 | N/A |
| pOPINB-LotA-N$^{1-300}$ (*L. pneumophila*) | Warren et al, 2023 | N/A |
| pOPINK-OTUD2$^{132-314}$ (*H. sapiens*) | Mevissen et al, 2013 | Addgene #61410 |
| pOPINS-TRABID$^{319-751}$ (*D. melanogaster*) | Xia et al, 2022 [preprint] | N/A |
| OTUB1* (*H. sapiens*) | Michel et al, 2015 | Addgene #65441 |
| AMSH* (*H. sapiens*) | Michel et al, 2015 | Addgene #66712 |
| pOPINB-Otulin (*H. sapiens*) | Keusekotten et al, 2013 | Addgene #61464 |
| pET28b-NUDT16 | Javed et al, 2023 | N/A |
| pET28b-MD1 | Javed et al, 2023 | N/A |
| pGEX-6p2-MD2 | Javed et al, 2023 | N/A |
| pGEX-6p2-TARG1 | Javed et al, 2023 | N/A |

| Reagent/resource | Reference or source | Identifier or catalog number |
|---|---|---|
| pCMV-GFP-PARP10 | Vyas et al, 2014 | Paul Chang at MIT (Cambridge, MA |
| pEGFP-C1 | Clontech | #6084-1 |
| pCW57-GFP-PARP10 | This study | N/A |
| pLP1, pLP2, pLP-VSVG | Gift from S. Jaffrey | N/A |
| HA-Ub | Gift from A. Barnes | N/A |
| **Antibodies** | | |
| Poly/Mono-ADP Ribose (E6F6A) (1:2000) | Cell signaling Technology | 83732 |
| Mono-ADP-Ribose AbD33204 (1:250 = 2 µg/ml) | Bio-Rad | HCA354 |
| Anti-poly-ADP-ribose binding reagent (1:1000) | Sigma-Aldrich | MABE1031 |
| GFP (1:2000) | Proteintech | pabg1 |
| Ubiquitin (P4D1) (1:1000) | Cell signaling Technology | 3936 |
| α-Tubulin (DM1A) (1:2000) | Cell signaling Technology | 3873 |
| HA-Tag (C29F4) (1:1000) | Cell signaling Technology | 3724 |
| anti-HA.11 Epitope Tag (1:2000) | Biolegend | 901501 (Previously Covance catalog# MMS-101P) |
| PARP1 (46D11) (1:2000) | Cell signaling Technology | 9532 |
| PARP10 (1:1000) | Abcam | ab70800 |
| PARP14 (1:1000) | Sigma-Aldrich | HPA012063 |
| Peroxidase AffiniPure™ Goat Anti-Rabbit IgG (H + L) (1:10,000) | Jackson ImmunoResearch | 111-035-144 |
| Goat anti-Mouse IgG (H + L) Secondary Antibody, HRP (1:5000) | Invitrogen | 62-6520 |
| **Chemicals, enzymes and other reagents** | | |
| DMEM, high glucose | Gibco | 11965118 |
| Fetal Bovine Serum | Sigma-Aldrich | F0926 |
| GlutaMAX™ Supplement | Gibco | 35050061 |
| Sodium Pyruvate (100 mM) | Gibco | 11360070 |
| Blasticidin S HCl | Corning | 30100RB |
| Doxycycline hyclate | Thermo Scientific Chemicals | 446060050 |
| jetOPTIMUS® DNA transfection Reagent | Polyplus-transfection | 101000025 |
| Calphos mammalian transfection kit | Takara | 631312 |
| Bovine Serum Albumin | Sigma-Aldrich | A9647 |
| Lenti-X™ GoStix™ Plus | Takara | 631280 |
| TCEP | Thermo Scientific | 20490 |
| cOmplete™, EDTA-free Protease Inhibitor Cocktail | Roche | 11873580001 |
| Phthal 01 (pan-PARP inhibitor) | Rodriguez et al, 2021 | N/A |

| Reagent/resource | Reference or source | Identifier or catalog number |
|---|---|---|
| PDD00017273 (PARG inhibitor) | Sigma-Aldrich | SML1781 |
| PR-619 | Medchemexpress | HY-13814 |
| Bio-Rad Protein Assay Dye Reagent Concentrate | Bio-Rad | 5000006 |
| 4–20% Mini-PROTEAN® TGX™ Precast Protein Gels | Bio-Rad | 4561096 |
| Nitrocellulose Transfer Kit | Bio-Rad | 1704270 |
| Carnation Instant Nonfat Dry Milk | Nestle | 12428935 |
| UltraPure™ Sodium Dodecyl Sulfate (SDS) | Invitrogen | 15525017 |
| SuperSignal™ West Pico PLUS Chemiluminescent Substrate | Thermo Scientific | 34578 |
| SuperSignal™ West Femto Maximum Sensitivity Substrate | Thermo Scientific | 34095 |
| ChromoTek GFP-Trap® Magnetic Agarose | Proteintech | gtma |
| Urea | Fisher Scientific | BP169 |
| Hydroxylamine solution | Sigma-Aldrich | 438227 |
| MG132 | Selleck Chemical | S2619 |
| Bortezomib | Millipore Sigma | 5.04314.0001 |
| TAK-243 | MedChemExpress | HY-100487 |
| RBN010860 | This study, synthesized as previous reported by Wigle et al, 2020 | |
| RBN012759 | Acme Biosciences | Custom synthesis |
| Adenosine 5′-diphosphoribose sodium salt (free ADPr) | Sigma-Aldrich | A0752 |
| β-NAD$^+$ | Sigma-Aldrich | N3014 |
| ATP | Sigma-Aldrich | A2383 |
| Benzonase | Millipore Sigma | 70664 |
| Recombinant Human IFN-beta | R&D Systems | 8499IF010 |
| Maltoheptaose | Sigma | M7753 |
| Silver Stain Plus Kit | Bio-Rad | 1610449 |
| HisPur Ni-NTA resin | Thermo Scientific | 88222 |
| **Software** | | |
| GraphPad Prism 10.1 | https://www.graphpad.com/ | |
| Image Lab | https://www.bio-rad.com/en-us/product/image-lab-software?ID=KRE6P5E8Z | |
| Unicorn | Cytiva | |
| **Other** | | |
| Trans-Blot® Turbo™ Transfer System | Bio-Rad | 1704150 |

| Reagent/resource | Reference or source | Identifier or catalog number |
|---|---|---|
| ChemiDoc MP imaging System | Bio-Rad | 12003154 |
| AKTA Pure | Cytiva | |
| Superdex75 Increase 10/300 GL | Cytiva | |

## Cell culture

All cells were cultured at 37 °C and 5% $CO_2$. HEK 293 control and PARP1 KO cells lines, a gift from Prof. Michael Garabedian at NYU Langone, were cultured in DMEM (Gibco,11965118) supplemented with 10% FBS (Sigma-Aldrich, F0926),1X GlutaMAX (Gibco, 35050061), and 1 mM sodium pyruvate (Gibco, 11360070). HEK 293 GFP-PARP10 doxycycline-inducible cells were cultured in DMEM (10% FBS, 1× GlutaMAX) supplemented with 10 µg/ml blasticidin (Corning, 30100RB) and treated with 10 µg/ml doxycycline (Thermo Scientific Chemicals, 446060050) for 24 h to induce GFP-PARP10 expression. For transient overexpression, cells were transfected with plasmids using jetOPTIMUS® DNA transfection Reagent (Polyplus-transfection, 101000025) for 24 h (media was exchanged 5 h post-transfection). All cell lines are routinely checked for mycoplasma contamination.

## Generation of GFP-PARP10 doxycycline-inducible cell line

HEK 293T (purchased from ATCC) cells were transfected with GFP-PARP10 (Vyas et al, 2014) (cloned into a pCW57 vector) and 3rd generation lentiviral packaging plasmids (pLP1, pLP2, pLP-VSVG), using Calphos mammalian transfection kit (Takara, 631312). Five hours after transfection, cells were washed once with media, and the transfection was continued overnight. After 24 and 48 h post-transfection, the virus-containing medium was collected, filtered through 0.45 µm cellulose acetate filter, and ultracentrifuged at 22,000 rpm for 2 h at 4 °C. The supernatant was discarded, and the virus-containing pellet was resuspended in 1% BSA in PBS (sterile-filtered through 0.45 µm cellulose acetate filter). The virus was quantified using Lenti-X™ GoStix™ Plus (Takara, 631280) and ~750 ng of virus was added to ~80% confluent HEK 293 PARP1 KO cells in a well of a six-well plate for transduction. Forty-eight hours post-transduction, cells were passaged using fresh media without virus, and 24 h later, cells were passaged again into media containing 10 µg/ml blasticidin for selection.

## Western blotting

Cells were lysed in cell lysis buffer (CLB; 50 mM HEPES pH 7.4, 150 mM NaCl, 1 mM $MgCl_2$, 1% Triton X-100) supplemented with fresh 1 mM TCEP, 1X cOmplete EDTA-free Protease Inhibitor Cocktail (Sigma-Aldrich, 11873580001) and 30 µM Phthal 01 (pan-PARP inhibitor) (Rodriguez et al, 2021). For Fig. EV1, 10 µM PDD00017273 was added to CLB, and for Figs. 3B, 4A, and 5A, 50 µM PR-619 was added to CLB. Lysates were centrifuged at 14,000 rpm for 10 min at 4 °C, the supernatant containing total protein was quantified with a Bradford assay (Bio-Rad,

5000006EDU). Sample buffer (10% glycerol, 50 mM Tris-Cl (pH 6.8), 2% SDS, 1% β-mercaptoethanol, 0.02% bromophenol blue) was added to normalized samples, followed by boiling at 95 °C for 5 min (or no boiling as indicated in figure legend) and resolving in either 10% SDS-PAGE gels or 4–20% precast gels (Bio-rad, 4561096). Proteins were transferred to a nitrocellulose membrane using Trans-Blot Turbo Transfer System (Bio-Rad), blocked in 5% milk (Carnation) in 1× PBS, 0.1% Tween-20 (PBST), and probed overnight at 4 °C with primary antibodies. The blots were washed three times with PBST and incubated for 1 h at room temperature with a goat anti-rabbit (1:10,000, Jackson ImmunoResearch Labs, 111035144) or goat anti-mouse (1:5000, Invitrogen, 62-6520) HRP-conjugated secondary antibody in 5% milk in PBST. After three washes with PBST, blots were developed with SuperSignal™ West Pico (Thermo Scientific, 34578) or Femto (Thermo Scientific, 34095) chemiluminescent substrate and imaged on a ChemiDoc Gel ImagingSystem (Bio-Rad).

A modified CLB was used in Fig. 5B and EV5C where, the 1% Triton X-100 was replaced with 1% SDS and 300 U/ml benzonase (Millipore Sigma, 70664), and lysates were not centrifuged or boiled after addition of sample buffer. These conditions were also used in Fig. 5C,F. However, lysates were diluted sixfold using standard CLB containing 1% Triton X-100, followed by the addition of TssM DUBs at 1 µM for 1 h at 37 °C.

## Immunoprecipitation and UbiCRest assay

GFP-Trap® Magnetic beads (Proteintech, gtma) were washed twice with CLB and added to lysates containing doxycycline-induced GFP-PARP10 (~300 µg total protein/10 µl bead slurry). The beads were placed on a rotator for 2 h at 4 °C, then washed once with CLB, twice with 7 M Urea/1% SDS in PBS, once with 1% SDS in PBS, and three times with HEPES buffer (Hb: 50 mM HEPES pH 7.5, 100 mM NaCl, 4 mM $MgCl_2$, 0.2 mM TCEP added fresh). In Fig. 3B, the beads were treated with TssM (2 µM), TssM* (2 µM), NUDT16 (10 µM, containing 15 mM $MgCl_2$), MD1 (2 µM), or hydroxylamine (1 M, in CLB pH 7.5) for 1 h at 37 °C, quenched with sample buffer and boiled at 95 °C for 5 min to elute samples for western blotting. In Fig. 4A, the beads were treated with NUDT16 (10 µM, containing 15 mM $MgCl_2$) for 1 h at 37 °C then the supernatant was removed, and the beads were washed once with Hb. Both supernatant and bead fractions were then further treated for 1 h at 37 °C with a panel of DUBs: TssM (2 µM), TssM* (2 µM), OUTLIN (1 µM), LotA-N (1 µM), Cezanne (0.2 µM), OTUD2 (1 µM), TRABID (0.2 µM), OTUB1* (1 µM), AMSH* (1 µM), USP21 (1 µM). The reaction was quenched with sample buffer and only the bead fraction was boiled at 95 °C for 5 min to elute GFP-PARP10 for detection by western blotting.

## Protein purification

The following proteins were expressed and purified as previously described: UBE1 (Gladkova et al, 2018), TssM/TssM* (Szczesna et al, 2024), OTULIN (Keusekotten et al, 2013), LotA-N/Ub (Warren et al, 2023), Cezanne (Mevissen et al, 2016), OTUD2 (Mevissen et al, 2013), TRABID (Xia et al, 2022), OTUB1*/AMSH* (Michel et al, 2015), USP21 (Ye et al, 2011). Full length ADPr hydrolases in a pET-28b vector (MD1, NUDT16) or in a pGEX-6p2 vector (MD2, TARG1) were expressed as previously reported

(Carter-O'Connell et al, 2014; Javed et al, 2023). UBE2D3 was expressed from a pET28a plasmid in Rosetta (DE3) cells. Cells were resuspended in 50 mM MES pH 6.0, 1 mM EDTA, sonicated to lyse, and the solution was clarified by centrifugation at 45000 g for 1 h. UBE2D3 was purified using a Resource S cation exchange chromatography column (Cytiva) and a gradient from 0 to 1 M NaCl. Fractions were pooled and concentrated for application to a HiLoad Superdex 75 pg 16/600 size exclusion column (Cytiva) equilibrated in 25 mM Tris, 150 mM NaCl, 2 mM DTT pH 7.4. Protein purity was evaluated by SDS-PAGE, and aliquots were stored at −80 °C until use. The RING-DTC domain of human DTX2 (residues 389–622) in a pET28b vector was transformed into *E. coli* Rosetta cells and grown overnight at 37 °C. A single colony was used to inoculate a starter culture of LB, which was then used to inoculate larger cultures for protein expression. At $OD_{600} = 0.6$, His-DTX2 expression was induced with 0.3 mM isopropyl β-d-1-thiogalactopyranoside (IPTG) overnight at 18 °C. The cells were harvested, resuspended in 50 mM HEPES pH 7.5, 500 mM NaCl, 10 mM imidazole, 2 mM β-mercaptoethanol, and lysed on ice by sonication. The lysate was clarified by centrifugation at 45000 g for 1 h, and the supernatant was applied to Ni-NTA resin for affinity purification. After binding, the resin was washed with 50 mM HEPES pH 7.5, 500 mM NaCl, 35 mM imidazole, 2 mM β-mercaptoethanol and the His-DTX2 protein was eluted in 50 mM HEPES pH 7.5, 500 mM NaCl, 250 mM imidazole, 2 mM β-mercaptoethanol. Eluted protein was dialyzed into 20 mM HEPES pH 7.5, 200 mM NaCl, 2 mM DTT overnight at 4 °C for application onto a Superdex 200 16/600 gel filtration column. Final purified His-DTX2 protein was snap frozen and stored at −80 °C until use.

### Ubiquitylation/deubiquitylation assay

Preparative amounts of E2~Ub conjugate was formed by reacting UBE1 (2 μM), UBE2D3 (100 μM), Ub (160 μM), MgCl$_2$ (20 mM), ATP (4 mM), and DTT (4 mM) in 25 mM HEPES, 150 mM NaCl pH 7.4 at 37 °C for 1 h. The reaction was then loaded onto a Superdex75 10/300 Increase column and protein species were separated using 25 mM HEPES, 150 mM NaCl pH 7.4. The eluted peak corresponding to the UBE2D3~Ub conjugate was collected, and the solution was concentrated to 500 μL. The prepared E2~Ub conjugate was then mixed with DTX2 (5 μM) and the appropriate substrate (NAD$^+$, ADPr; 0.2 mM) for 30 min at 37 °C. These reactions were then diluted 2:1 for the deubiquitylation experiment. Samples were treated with TssM or TssM* (0.5 μM), NUDT16 (10 μM, with 15 mM MgCl$_2$) or NH$_2$OH (1 M in HEPES buffer pH 7.4) for 30 min at 37 °C. All reactions in the deubiquitylation experiment contained 5 mM DTT. Gel samples taken over the whole experiment were quenched in 3× sample buffer, separated on 4–20% Mini-PROTEAN TGX gels (Bio-Rad Laboratories) and silver stained. Maltoheptaose-Ub (Malt-Ub), prepared as previously described (Kelsall et al, 2022), was incubated with 2 μM TssM, 2 μM TssM* or 1 M NH$_2$OH (in HEPES pH 7.4 buffer) for 1 h at 37 °C, then quenched with sample buffer, followed by western blotting.

## Data availability

This study includes no data deposited in external repositories.

The source data of this paper are collected in the following database record: biostudies:S-SCDT-10_1038-S44318-025-00391-7.

## Peer review information

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

## Acknowledgements

The authors thank current and former Cohen and Pruneda lab members for many insightful discussions relating to experimental design, data analysis and interpretation, and general advice and feedback for this project. The authors thank Dr. Sunil Sundalam (former member of Cohen lab) for synthesizing RBN010860. The authors thank Ivan R Siordia for expressing and purifying ADPr hydrolase recombinant proteins used in this study. This work was supported by the National Institutes of Neurological Disorders and Stroke 2R01NS088629 (to MSC) and by the National Institute of General Medical Sciences funding grant R35GM142486 (to JNP). Additional support was

provided by Achievement Rewards for College Scientists (ARCS) and a National Cancer Institute NRSA F31 fellowship F31CA284712 (to DSB).

## Author contributions

**Daniel S Bejan**: Conceptualization; Validation; Investigation; Visualization; Methodology; Writing—original draft; Writing—review and editing. **Rachel E Lacoursiere**: Resources; Investigation; Writing—review and editing. **Jonathan N Pruneda**: Conceptualization; Resources; Funding acquisition; Writing—review and editing. **Michael S Cohen**: Conceptualization; Supervision; Funding acquisition; Writing—review and editing.

Source data underlying figure panels in this paper may have individual authorship assigned. Where available, figure panel/source data authorship is listed in the following database record: biostudies:S-SCDT-10_1038-S44318-025-00391-7.

## Disclosure and competing interests statement

The authors declare no competing interests.

# Expanded View Figures

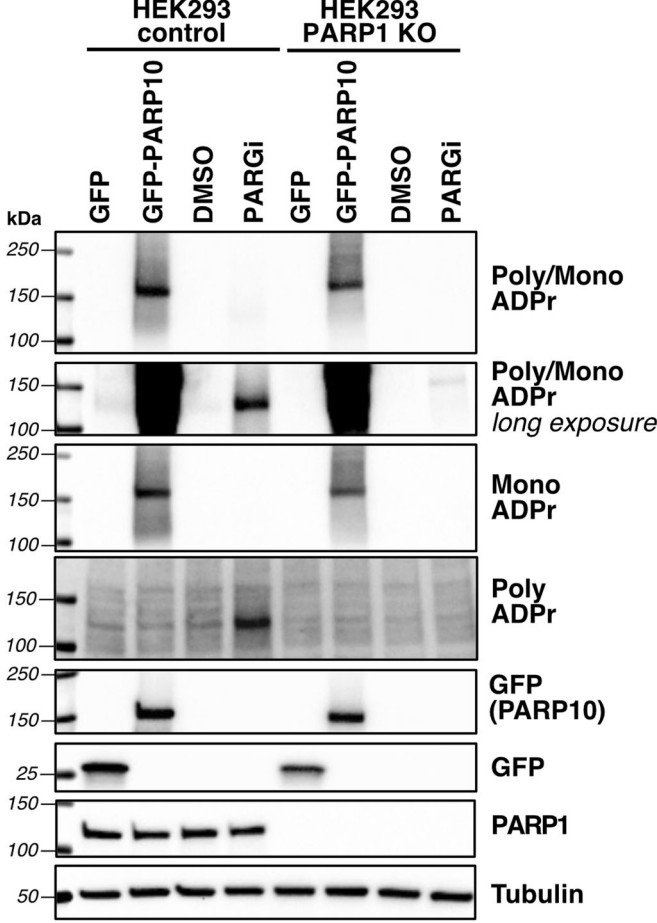

**Figure EV1. Evaluating specificity of poly-ADPr, mono-ADPr, and poly/mono-ADPr antibodies.**

HEK 293 control and PARP1 KO cells were transiently transfected with GFP or GFP-PARP10 for 24 h, or treated with PARG inhibitor (PDD00017273, 1 μM) for 30 min, followed by western blotting and probing for poly/mono ADPr (Cell Signaling Technology: E6F6A), mono-ADPr (Bio-Rad: HCA354), or poly-ADPr (Millipore Sigma: MABE1031). Representative western blot from $n = 2$ biological replicates. Source data are available online for this figure.

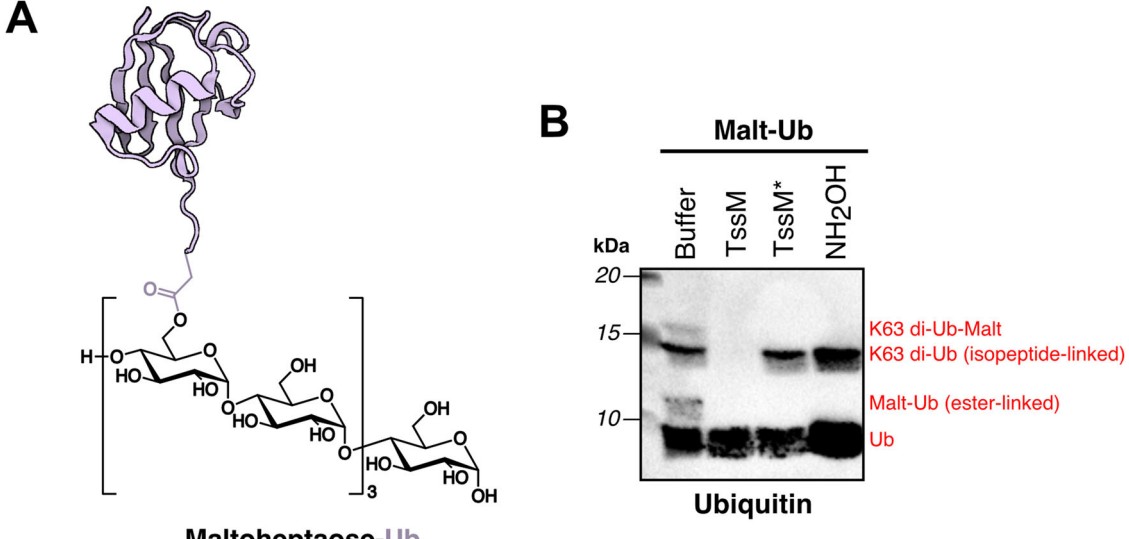

**Figure EV2. TssM\* is highly specific for removal of ester-linked maltoheptaose-Ub.**

(A) Structure of maltoheptaose-Ub (Malt-Ub) with an ester-linked Ub on the C6 hydroxyl group of glucose. (B) Malt-Ub was incubated with 2 μM TssM, 2 μM TssM\*, or 1 M NH$_2$OH (pH 7.5) for 1 h at 37 °C. Representative western blot from $n = 3$ biological replicates. Source data are available online for this figure.

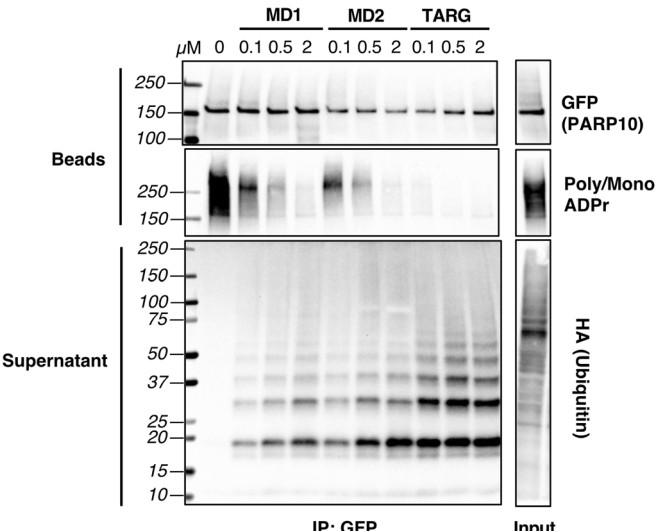

**Figure EV3.  Screening macrodomain-based ADPr hydrolases against PARP10 MARylation.**

GFP-PARP10, from doxycycline-induced HEK 293 cells transfected with HA-Ub, was immunoprecipitated with GFP-trap beads, washed stringently (7 M Urea, 1% SDS), and beads were treated with a dose-response of MacroD1 (MD1), MacroD2 (MD2) or terminal ADP-Ribose protein glycohydrolase (TARG). The supernatant and bead fractions were separated and subjected to western blotting. Representative western blot from $n = 2$ biological replicates. Source data are available online for this figure.

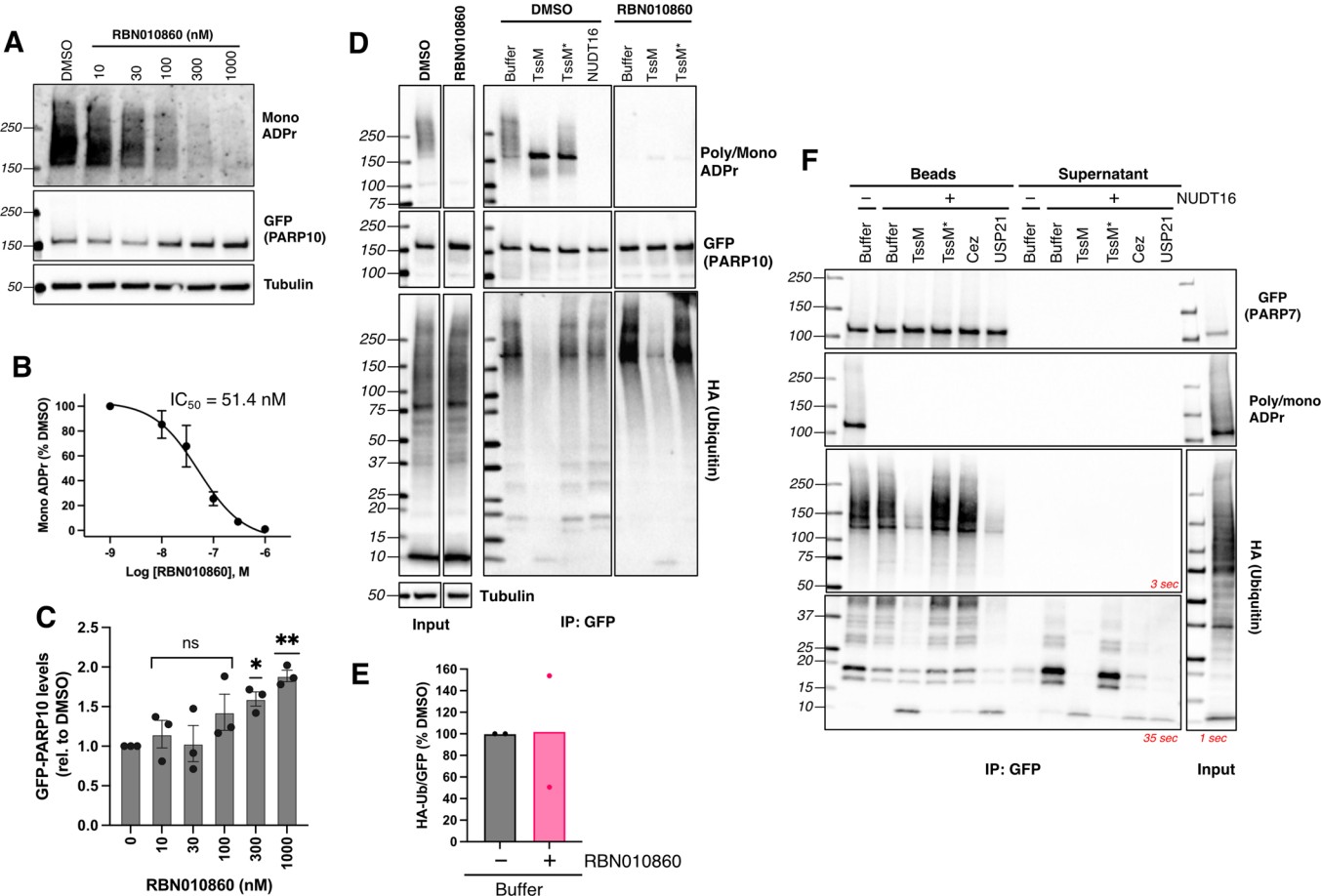

**Figure EV4.  PARP10 contains isopeptide-linked polyUb that is not dependent on MARylation, and PARP7 contains K11-linked MARUbylation.**

(A) HEK 293 cells with doxycycline-inducible GFP-PARP10 were co-treated with doxycycline (10 μg/ml) and a dose-response of RBN010860 for 24 h, followed by western blotting. Quantification of the IC$_{50}$ value of RBN010860 (B) and GFP-PARP10 levels (C) from (A); $n = 3$ biological replicates. Asterisks indicate statistical significance (ns: not significant). For (C), 300 nM RBN010860: *$P = 0.0219$, 1000 nM RBN010860: **$P = 0.0064$, calculated using a one-sample t and Wilcoxon test relative to vehicle treatment. (D) GFP-PARP10 dox-inducible HEK 293 cells were transfected with HA-Ub and treated with DMSO or 1 μM RBN010860 for 24 h, followed by immunoprecipitation with GFP-trap beads, stringent washing (7 M Urea, 1% SDS), and on-bead treatment with TssM (1 μM) or NUDT16 (10 μM). (E) Quantification of the HA-Ub signal (normalized to GFP) from buffer-treated samples from DMSO or RBN010860 treatment; $n = 2$ biological replicates. (F) GFP-PARP7 and HA-Ub were transiently co-expressed in HEK 293 T cells and immunoprecipitated with GFP-trap beads, followed by stringent washing (7 M Urea, 1% SDS), and treatment with NUDT16. The beads and supernatant were separated and further treated with 2 μM TssM, 2 μM TssM*, 0.2 μM Cezanne, or 1 μM USP21, followed by western blotting (representative image of $n = 1$ biological replicate). Source data are available online for this figure.

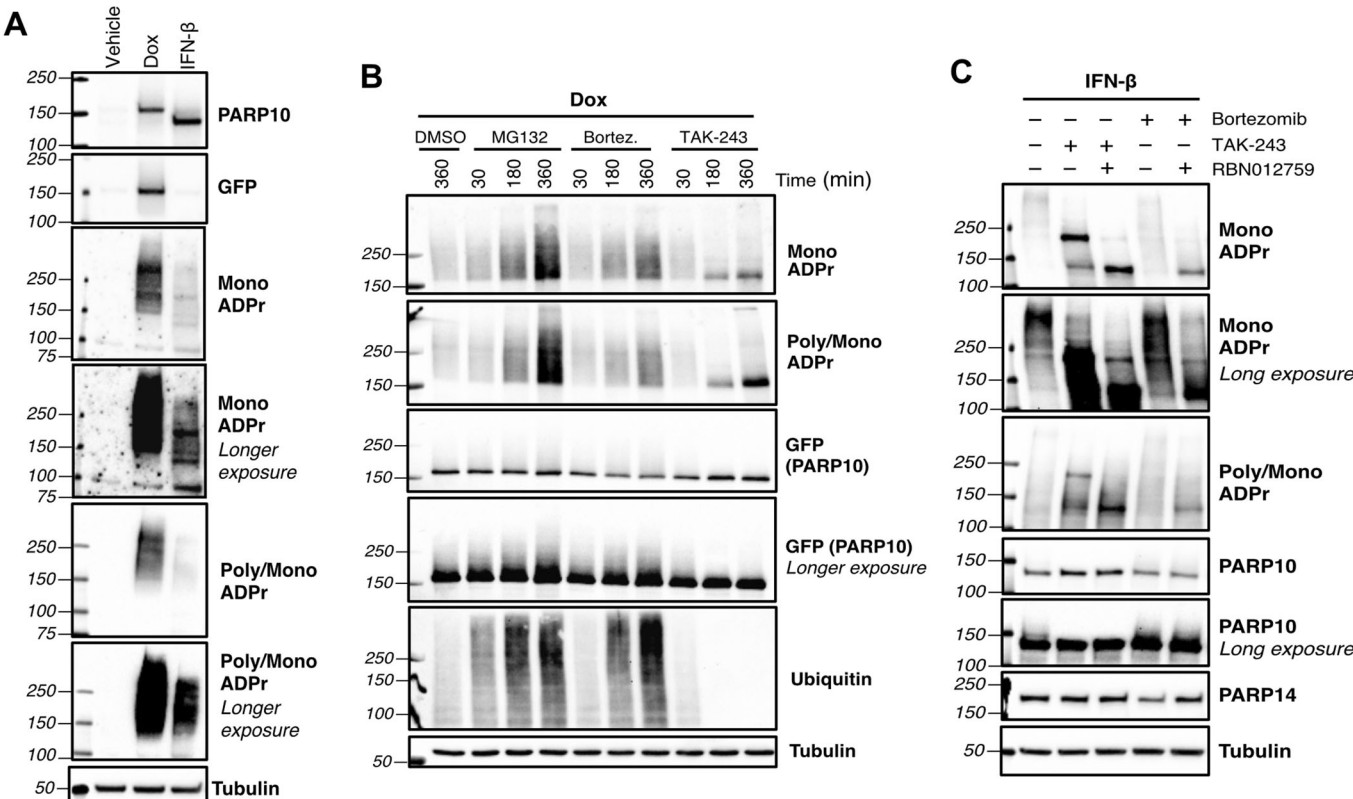

**Figure EV5.    Inhibition of the ubiquitin-proteasome pathway alters dox-inducible GFP-PARP10 and IFN-β-inducible MARUbylation.**

(A) HEK 293 GFP-PARP10 doxycycline-inducible cells were treated with doxycycline (10 μg/ml) or IFN-β (140 U/ml) for 24 h, followed by western blotting (n = 1 biological replicate). (B) HEK 293 GFP-PARP10 doxycycline-inducible cells were treated with doxycycline (10 μg/ml) for 24 h, followed by a time course with MG132 (10 μM), bortezomib (0.1 μM), and TAK-243 (1 μM). Cells were lysed in CLB followed by western blotting. Representative western blot from n = 3 biological replicates. (C) PARP1 KO HEK 293 cells were treated with IFN-β (140 U/ml) and RBN012759 (0.1 μM) for 24 h, followed by 3-h treatment with MG132 (10 μM) or bortezomib (0.1 μM). Cells were lysed in CLB with 1% SDS followed by western blotting (no boiling of samples). Representative western blot from n = 2 biological replicates. Source data are available online for this figure.

