## [Peer Review File · The EMBO Journal]

Ubiquitin is directly linked via an ester to protein-conjugated mono-ADP-ribose

Daniel Bejan, Rachel Lacoursiere, Jonathan Pruneda, and Michael Cohen

Corresponding author(s): Michael Cohen (cohenmic@ohsu.edu), Jonathan Pruneda (pruneda@ohsu.edu)

Review Timeline:

Submission Date:	13th Jul 24
Editorial Decision:	9th Aug 24
Revision Received:	24th Dec 24
Editorial Decision:	24th Jan 25
Revision Received:	1st Feb 25
Accepted:	7th Feb 25

Editor: Hartmut Vodermaier

Transaction Report:

Prof. Michael S Cohen
Oregon Health and Science University, Portland, OR, USA
Chemical Physiology and Biochemistry

9th Aug 2024

Re: EMBOJ-2024-118401
Mono-ADP-ribosylation is directly ester-linked to polyubiquitin in cells

Dear Dr. Cohen,

Thank you again for submitting your study on PARP10 and links between cellular ADP-ribosylation and ubiquitylation to The EMBO Journal. We have now received the reports of three expert referees, copied below for your information. As you will see, the referees find the results potentially interesting and appreciate the clear biochemical demonstration of the link between the two modifications. However, they remain to varying degrees unconvinced that the physiological relevance of this phenomenon is already sufficiently supported; in particular regarding the dependency on PARP10 expression levels, and the effects on PARP10 stability/proteasomal degradation, points that we feel would be essential to follow up on in order to warrant publication in a broad general journal such as this one. In addition, the reviewers also list a variety of more specific/more minor issues that would need to be addressed.

In this situation, I remain open to giving you an opportunity to respond to the reports by way of a revised version of the manuscript, but should emphasize that we can only pursue the study further for The EMBO Journal in case you should be able to decisively strengthen the evidence for normal physiological occurrence and significance of the double modification in cells, as this would be the major advance in the study. I would therefore encourage you to contact me with a revision plan and preliminary point-by-point response already during the early stages of your revision work, in order to clarify/discuss if and how key issues raised in the reports may be solved. We would also be open to extension of the default three-months revision period if needed; our 'scooping protection' (meaning that competing work appearing elsewhere in the meantime will not affect our considerations of your study) would of course remain valid also throughout such an extension.

Further information on preparing, formatting and uploading a revised manuscript can be found below and in our Guide to Authors. Thank you again for the opportunity to consider this work for The EMBO Journal, and I look forward to hearing from you in due time.

With kind regards,

Hartmut

9) To facilitate reproducibility and cross-laboratory adoption of methodologies, please structure the Materials & Methods section as outlined in our guide to authors, including a completed Reagents and Tools Table that can be downloaded from our author guidelines as well (<https://www.embopress.org/page/journal/14602075/authorguide#structuredmethods>).

10) Digital image enhancement is acceptable practice, as long as it accurately represents the original data and conforms to community standards. If a figure has been subjected to significant electronic manipulation, this must be clearly noted in the figure legend and/or the 'Materials and Methods' section. The editors reserve the right to request original versions of figures and the original images that were used to assemble the figure. Finally, we generally encourage uploading of numerical as well as gel/blot image source data; for details see: embopress.org/page/journal/14602075/authorguide#sourcedata

At EMBO Press, we ask authors to provide source data for the main manuscript figures. Our source data coordinator will contact you to discuss which figure panels we would need source data for and will also provide you with helpful tips on how to upload and organize the files.

Revision to The EMBO Journal should be submitted online within 90 days, unless an extension has been requested and approved by the editor; please use the link below to submit the revision online before 7th Nov 2024:

Link Not Available

If you choose to alternatively have this study further considered by another EMBO Press publication, you may use the following hyperlink to transfer the manuscript, optionally with inclusion of referee reports and identities:

Link Not Available

Referee #1:

This study by the Cohen lab presents *in vivo* evidence for ester-linked mono-ADP-ribose poly-ubiquitination on overexpressed forms of the PARP10 and PARP7 MARYlases. Using a clever combination of chemical and enzymatic reagents, they indirectly show that ester-linked ubiquitination of ADP-ribose deposited on glutamate/aspartate residues is present on these MARYlases, demonstrating the existence of a new post-translational modification in mammalian cells. Ubiquitination of ADP-ribose deposited on proteins and nucleic acids has been recently shown to be carried out *in vitro* by the DELTEX family of E3 ubiquitin ligases (Zhu et al. *Sci Adv* 2022, Zhu et al. *NAR* 2024) and the demonstration that such a modification does occur *in vivo* is an important step forward in the field.

They also propose that addition of K11-linked ubiquitin chains onto ADP-ribosylated PARP10 favors its proteasomal degradation but the data to support this model are not very conclusive. To make the manuscript suitable for publication will require

addressing a few important points.

Major points :

1. In figure 5, the authors show that MG132 enhances the poly/mono ADPr signal in overexpressed PARP10 and an upper band in GFP-PARP10 which is thought to be a MARylated /ubiquitinated form of PARP10 as this band is prevented by treatment with the E1 inhibitor TAK-243. In B a quantification of 2 biological replicates is also shown but does not support an increase in the upper PARP10 band in the MG132-treated samples. This experiment should be repeated, quantified and analyzed to provide statistical support for the conclusion. Additionally, MG132 is a broad protease inhibitor. More conclusive evidence for the proteasomal degradation of MARubylated PARP10 should be obtained using a more specific proteasome inhibitor such as bortezomib.
2. Similarly, in 5C and D, cycloheximide experiments are performed to show that inhibition of the MARylase activity of PARP10 enhances its stability yet the quantification of the 2 replicates does not clearly demonstrate this. Additional repeats should be performed. The same critique can be made for figure S4C.

Minor point :

1. The RBN010860 inhibitor targets several PARP enzymes apart from PARP10 itself. Could a catalytically-dead mutant be used to demonstrate for instance a decrease in MARylation, ubiquitination and an overall increase in PARP10 stability as expected from the model ?

Typos and grammar

Page 2 were reported to ubiquitylate a hydroxyl group...

Page 6 Previous studies demonstrated that acidic amino acids...

Page 8 As an orthogonal strategy, we showed that hydroxylamine...

Page 16 and the transfection was continued overnight.

Page 16 For Figure 1S (likely a mislabeling error)

Referee #2:

This manuscript by Bejan et al describes an interesting set of results suggesting that PARP10 is modified in cells both with canonical isopeptide-linked ubiquitin as well as with mono-ADPr-linked poly-ubiquitin chains, which they term "MARUbylation". Using elegant chemical and enzymatic methods to process these different PARP10 modifications, the authors argue that both the ADPr and the Ub within the "MARUbe" are ester-linked, and that the poly-ubiquitin chain is composed of K11 linkages. These are highly relevant and timely findings, as ester-linked ubiquitination is of growing interest, and ADPr-linked ubiquitin had only been demonstrated in biochemical reactions so far. However, the concerns listed below should be adequately addressed in a revised version of the manuscript.

Major concerns:

1. The authors mention in the discussion that the detection of the reported PARP10 modifications depends on PARP10 expression levels (Fig. 1A vs Fig. S1). Can the authors detect any of the modifications shown in the manuscript on endogenous PARP10 protein or are they only observed when PARP10 is overexpressed? How do the doxycycline-induced PARP10 expression levels in their system compare to endogenous PARP10 protein levels?
2. The initial characterization of the TssM and TssM* enzymes, which are used to distinguish between isopeptide and ester-linked ubiquitin modifications throughout the manuscript, relies on two substrates, DTB-NAD⁺-Ube (Fig. 2) and maltoheptaose-Ub (Fig. S2), neither of which is adequately characterized. The authors should demonstrate that incubation of DTX2 with DTB-NAD⁺ and the E2-Ub complex indeed generates the molecule proposed in Fig. 2A. Considering recent publications in the field (Zhu et al Sci Adv 2022), this is likely to be the case, but the authors should confirm this observation to strengthen this idea. Regarding the maltoheptaose-Ub substrate, I could not find the description of how this substrate was generated, and the authors should include data to support that the bands in Fig. S2B indeed represent the named species.
3. There are small molecular weight HA-ubiquitin bands in the GFP-PARP10 IP bead samples (Fig. 3B, 4A and S5, 15-50 kDa range). Given that the IP is affinity-purifying GFP-PARP10 (>150kDa), and the IP is performed under stringent (7M Urea) conditions, the appearance of these bands is surprising. It is interesting to note that this signal is not seen with RBN010860 treatment (Fig. S5A, HA-ubiquitin blot, 15-50kDa range) and these bands appear slightly bigger in the NUDT16, MD1 and hydroxylamine samples than in the TssM* samples (Fig. 2B, HA-ubiquitin blot, 15-50kDa range). The authors should discuss possible explanations for the origin of these bands.
4. There is no data or indeed any discussion of the E3 ligases that could catalyse the isopeptide-linked ubiquitination or the MAR-linked K11 poly-Ub chain modification on PARP10. Ideally, the authors could provide data testing the effect of depleting some candidate E3 ligases, but failing that, at least a discussion of some candidates is necessary.
5. Can the authors demonstrate that depleting Cezanne, the K11 poly-Ub deubiquitinase identified in the UbiCREST assay, or

any other K11 deubiquitinase, affects PARP10 modification or PARP10 protein levels?

Minor concerns:

6. Is the PARP10 auto-MARylation signal the only ADPr signal induced with GFP-PARP10 expression? An image of a full molecular weight range ADPr blot would resolve this.
7. Can ADPr detection reagents detect an ADPr moiety on the released di-Ub species in the IP supernatants (Fig. S3 and 4A)?
8. There are additional bands in the 50-75 kDa range detected using the Ub(FK2) antibody in Fig. 2B, only in the Tssm* sample. Can the authors determine the origin of these species?
9. The blots in Fig. S5 are substantially spliced, making it difficult to properly compare signal intensities between samples. Based on other data in the manuscript (Fig. 3B), wouldn't the authors expect RBN010860 to reduce the GFP-PARP10 ubiquitination to around ~60% of the DMSO control?
10. Fig. 4C shows that Cezanne is the only DUB within the panel that can hydrolyse K11 linkages, but this experiment alone is insufficient to demonstrate that Cezanne is strictly specific for K11, as other linkages were not tested. I suggest revising the text accordingly.

Errors/typos:

pg. 5-bottom: "doxycycline (dox)-inducible HEK 293 cells" is imprecise. It is the GFP-PARP10 expression that is dox-induced, not the cells. There are other similar examples, I suggest careful revision.

Fig 2B: indicate the likely TssM bands next to the picture of the coomassie-stained gel

Pg.8: correct the sentence to: "DTX2 was also reported to catalyze isopeptide-linked ..."

Referee #3:

The manuscript describes an investigation into the molecular action of PARPs and their ability to transfer ADP-ribose onto protein substrates. The study is mostly focused on PARP10 and its ability to auto-modify itself. Evidence is provided that the ADP-ribose moiety is further modified with poly-ubiquitin chains, though it remains possible that ubiquitylation occurs during lysate preparation and not within the normal context of cellular function. PARP10 was also shown to be directly modified with poly-ubiquitin chains (eg. auto-ubiquitylation), though the significance of this was unclear. Chain linkage analysis revealed that ubiquitin chains attached to ADP-ribose were formed through Lys 11. Experiments were performed in the presence of MG-132, a well-known proteasome inhibitor, or TAK-243, an inhibitor of ubiquitylation. MG-132 treatment resulted in the appearance of a band migrating above GFP-PARP10 and apparent stabilization of the ADP-ribosylated form (but no apparent changes in the levels of unmodified GFP-PARP10). Cycloheximide chase experiments demonstrated that GFP-PARP10 protein levels are unstable in a manner that depended on PARP10 enzymatic activity.

Taken together, the findings are convincing that a poly-ubiquitin chain may be established onto the ADP-ribose moiety on auto-modified GFP-PARP10. However, the physiological relevance of this modification is dubious. While changes occur to GFP-PARP10 upon treatment with proteasome inhibitor, there is no clear effect on GFP-PARP10 levels (Figure 5A). Furthermore, the degree of modification on GFP-PARP10 appears to be very subtle (Figure 3B). While this latter point was acknowledged by the authors in the Discussion section (and attention was drawn to the fact that ester bonds are labile), this still doesn't explain the lack of PARP10 stabilization upon treatment of cells with MG-132. As such, in the reviewer's opinion, the authors convincingly demonstrate that while dual modification of PARP10 by ADP-ribose and a poly-ubiquitin chain is possible in principle, it is still very much an open question whether this is somehow an artifact of the experiment or of bona fide biological relevance.

Comments

Page 6, last line. The authors claim that susceptibility of the sample to hydroxylamine is evidence that MARylation occurs on Glu/Asp residues. How can they rule out that the modification is not occurring on Ser residues, since oxyester linkages to Ser may also be hydrolyzed by hydroxylamine? It was also stated that "we treated lysates with neutral hydroxylamine", but this is likely not accurate since the pKa of hydroxylamine is approximately 6.

Figure 2B. The authors claim that DTX2 auto-ubiquitylation occurs on Lys residues which appears consistent with the anti-Ubiquitin FK2 Ab Western blot. Laddering is observed when DTX is added to the reaction but then disappears upon treatment with the DUB TssM. What's unclear is why the laddering is more pronounced and the intensity of the bands so much greater in the sample treated with TssM*, a mutant variant that is supposedly far more specific as an esterase compared with isopeptidase activity.

Figure 3B. There appears to be no negative control for the immuno-precipitation such as GFP alone. This significantly weakens the interpretations made by the authors.

Manuscript number: EMBOJ-2024-118401

Response to Reviewer Comments

We thank the reviewers for their interest and thoughtful commentary on our manuscript. Their insights and constructive criticisms have been instrumental in preparing a revised version of this manuscript.

The revised manuscript contains the following significant additions: (1) demonstration that MARUbylation occurs on endogenous PARPs in cells upon type I interferon stimulation and (2) evidence that MARUbylation occurs in cells and not in lysates.

Detailed responses to reviewers are provided below:

Referee #1:

“This study by the Cohen lab presents in vivo evidence for ester-linked mono-ADP-ribose poly-ubiquitination on overexpressed forms of the PARP10 and PARP7 MARYlases. Using a clever combination of chemical and enzymatic reagents, they indirectly show that ester-linked ubiquitination of ADP-ribose deposited on glutamate/aspartate residues is present on these MARYlases, demonstrating the existence of a new post-translational modification in mammalian cells. Ubiquitination of ADP-ribose deposited on proteins and nucleic acids has been recently shown to be carried out in vitro by the DELTEX family of E3 ubiquitin ligases (Zhu et al. Sci Adv 2022, Zhu et al. NAR 2024) and the demonstration that such a modification does occur in vivo is an important step forward in the field.

They also propose that addition of K11-linked ubiquitin chains onto ADP-ribosylated PARP10 favors its proteasomal degradation but the data to support this model are not very conclusive. To make the manuscript suitable for publication will require addressing a few important points.”

Major points:

1. “In figure 5, the authors show that MG132 enhances the poly/mono ADPr signal in overexpressed PARP10 and an upper band in GFP-PARP10 which is thought to be a MARYlated /ubiquitinated form of PARP10 as this band is prevented by treatment with the E1 inhibitor TAK-243. In B a quantification of 2 biological replicates is also shown but does not support an increase in the upper PARP10 band in the MG132-treated samples. This experiment should be repeated, quantified and analyzed to provide statistical support for the conclusion. Additionally, MG132 is a broad protease inhibitor. More conclusive evidence for the proteasomal degradation of MARubylated PARP10 should be obtained using a more specific proteasome inhibitor such as bortezomib.”

Response: The referee raises an important point that we acknowledge needs to be addressed. We have repeated this experiment several more times and have been unable to observe the distinct upper GFP-PARP10 band consistently. Due to the lack of reproducibility of this result, we felt it was best to omit it and the claim that MARUbylation leads to proteasomal degradation from our manuscript (previous Fig. 5).

Although we did not consistently observe an increase in the upper GFP-PARP10 band, we repeatedly observed a high (>150 kDa) molecular weight (MW) GFP-PARP10 and corresponding MARYlation smear following treatment with MG132 for progressively longer durations (new Fig. EV5B). As suggested by the reviewer, we treated cells with the structurally distinct and more selective proteasome inhibitor, bortezomib, and observed a similar effect (new

Fig. EV5B). The underlying cause of the increased MARUbylation upon proteasomal inhibition is unclear. One possibility is that proteasome inhibition stabilizes a protein that negatively regulates an ADPr hydrolase or DUB, resulting in an accumulation of MARUbylation. We discuss this possibility in the revised manuscript. Interestingly, we observe a similar phenomenon with endogenous PARP10 using IFN- β (new **Figure EV5C**).

2. 'Similarly, in 5C and D, cycloheximide experiments are performed to show that inhibition of the MARYlase activity of PARP10 enhances its stability yet the quantification of the 2 replicates does not clearly demonstrate this. Additional repeats should be performed. The same critique can be made for figure S4C.'

Response: As recommended by the referee, we repeated the cycloheximide time course experiment and included additional time points (see **Fig. 1R** below). While inhibition of PARP10 activity (1 μ M RBN010860 overnight pre-treatment) in the presence of cycloheximide showed a trend toward increased GFP levels, the difference was not statistically significant. This suggests, perhaps, that the canonical linked polyUb species on PARP10 contains a degradation signal (e.g., K48), and that it is not dependent on PARP10 catalytic activity. Separating the outcomes of the canonical polyUb and the MARUbylation marks, therefore, is highly complex and awaits future discoveries in this system We have removed previous **Fig. 5C,D**.

Fig. 1R. Catalytic activity does not appear to regulate PARP10 stability in cells. GFP-PARP10 dox-inducible HEK293 cells were treated with dox (10 μ g/ml) +/- RBN010860 (1 μ M) for 24 hr, followed by a time course with CHX (10 μ g/ml).

Regarding previous **Fig. S4C**, we have repeated this experiment one more time for an $n = 3$ total and show a statistically significant increase in GFP-PARP10 levels upon treatment with 300 and 1000 nM RBN010860 (see new **Fig. EV4A-C**). Given the results in new **Fig. EV5**, we surmise that the increase in PARP10 levels is due to the collapse of high MW, MARUbylated GFP-PARP10.

Minor points:

1. "The RBN010860 inhibitor targets several PARP enzymes apart from PARP10 itself. Could a catalytically-dead mutant be used to demonstrate for instance a decrease in MARYlation, ubiquitination and an overall increase in PARP10 stability as expected from the model?"

Response: This is a good suggestion by the referee, but unfortunately, we do not have a stable, dox-inducible cell line expressing catalytically dead GFP-PARP10. However, given our recent results, we no longer claim a model where PARP10 catalytic activity drives PARP10 degradation.

Typos and grammar

Page 2 were reported to ubiquitylate a hydroxyl group...

Page 6 Previous studies demonstrated that acidic amino acids...

Page 8 As an orthogonal strategy, we showed that hydroxylamine... [deleted]

Page 16 and the transfection was continued overnight.

Page 16 For Figure 1S (likely a mislabeling error)

The aforementioned typos and grammar have been addressed. The "Page 8" statement has been deleted and replaced with new text that goes along with a new figure.

Referee #2:

"This manuscript by Bejan et al describes an interesting set of results suggesting that PARP10 is modified in cells both with canonical isopeptide-linked ubiquitin as well as with mono-ADPr-linked poly-ubiquitin chains, which they term "MARUbylation". Using elegant chemical and enzymatic methods to process these different PARP10 modifications, the authors argue that both the ADPr and the Ub within the "MARUbe" are ester-linked, and that the poly-ubiquitin chain is composed of K11 linkages. These are highly relevant and timely findings, as ester-linked ubiquitination is of growing interest, and ADPr-linked ubiquitin had only been demonstrated in biochemical reactions so far. However, the concerns listed below should be adequately addressed in a revised version of the manuscript."

Major concerns:

1. "The authors mention in the discussion that the detection of the reported PARP10 modifications depends on PARP10 expression levels (Fig. 1A vs Fig. S1). Can the authors detect any of the modifications shown in the manuscript on endogenous PARP10 protein or are they only observed when PARP10 is overexpressed? How do the doxycycline-induced PARP10 expression levels in their system compare to endogenous PARP10 protein levels?"

Response: We value the referee's point, which encouraged us to explore whether we could detect endogenous MARYlation. However, detecting endogenous ADP-ribosylation is challenging, especially for MARYlation, due to the low stoichiometry of the modification and low abundance (single or double-digit nanomolar concentrations in HEK 293 cells, <https://opencell.czbiohub.org/>) of MARYlating PARPs. Even for the well-studied, abundant (micromolar concentrations in HEK 293 cells, <https://opencell.czbiohub.org/>) PARP1, endogenous ADP-ribosylation can only be detected when a stimulus is added (e.g., H₂O₂-mediated DNA damage) or if the ADPr hydrolases (i.e., PARG and ARH1) are knocked out or inhibited. To our knowledge, there are currently no reported studies detecting endogenous PARP10 ADP-ribosylation activity. The stimulus/cellular context required to observe endogenous MARYlation mediated by PARP10 or other PARP family members is poorly understood. Only recently, it has been shown that the type II interferon, IFN- γ , can induce

PARP14-mediated auto-MARylation and MARylation of other targets in several cell lines (e.g., A549) [Kar et al., EMBO 43:2929 (2024)]. Although detecting endogenous MARylation is challenging, this result motivated us to ask if the type I, IFN- β , which is known to induce PARP10 expression (<https://interferome.org/interferome/>), could induce PARP10 expression and MARylation in HEK 293 cells.

Gratifyingly, we found that treatment with IFN- β led to the detection of endogenous PARP10 (as well as PARP14), and a MARylation smear originating at the molecular weight of PARP10 (new Fig. 5A). The expression level of IFN- β -induced endogenous PARP10 was similar to that of dox-inducible GFP-PARP10 (new Fig. EV5A). PARP inhibitor studies and treatment with TssM and TssM* provide compelling evidence of endogenous MARUbylation on PARP10 and PARP14 (new Fig. 5B, F-H).

2. “The initial characterization of the TssM and TssM* enzymes, which are used to distinguish between isopeptide and ester-linked ubiquitin modifications throughout the manuscript, relies on two substrates, DTB-NAD⁺-Ube (Fig. 2) and maltoheptaose-Ub (Fig. S2), neither of which is adequately characterized. The authors should demonstrate that incubation of DTX2 with DTB-NAD⁺ and the E2-Ub complex indeed generates the molecule proposed in Fig. 2A. Considering recent publications in the field (Zhu et al Sci Adv 2022), this is likely to be the case, but the authors should confirm this observation to strengthen this idea. Regarding the maltoheptaose - Ub substrate, I could not find the description of how this substrate was generated, and the authors should include data to support that the bands in Fig. S2B indeed represent the named species.”

Response: We agree with the referee that additional characterization is needed for the DTB-NAD-Ube substrate generated in Fig. 2. As an initial test, we incubated DTB-NAD-Ube generated by DTX2, with NUDT16, and observed a significant decrease in the DTB-NAD-Ube signal (Fig. 2R). This surprising result suggests that Ub could be attached to a hydroxyl group of the nicotinamide-proximal ribose, or perhaps at the canonical C1' position, generating ADP-ribosylation (as proposed by Ref 16,17).

Fig. 2R. NUDT16 reduces the DTB-NAD-MARUbe signal. DTX2 was incubated with 25 μ M DTB-NAD⁺ and a loaded E2-Ub complex (de-salted) for 1 hr at 37°C, followed by 1 hr reaction at 37°C with 1 μ M TssM, 1 μ M TssM*, 1M NH₂OH (pH 7.5), or 10 μ M NUDT16.

A potential explanation for this unexpected result could be using a modified (i.e., desthiobiotin-modified) NAD⁺ analog instead of native NAD⁺. Therefore, we replaced DTB-NAD⁺ with native NAD⁺ or ADPr (as done in the Zhu et al. paper) to generate the ester-linked species. We now show that treatment with NUDT16 no longer cleaves the NAD-Ub or ADPr-Ub signal, while TssM, TssM, and hydroxylamine do (new Fig. 2).*

Regarding the Malt-Ub substrate (previous Fig. S2; now Fig. EV2), this was generated as described in Kelsall et al. 2022. The reaction conditions produce a characteristic mass-shifted product for the Malt-Ub product. We have included the description of Malt-Ub generation in the methods section and cited Kelsall et al. 2022

3. “There are small molecular weight HA-ubiquitin bands in the GFP-PARP10 IP bead samples (Fig. 3B, 4A and S5, 15-50 kDa range). Given that the IP is affinity-purifying GFP-PARP10 (>150kDa), and the IP is performed under stringent (7M Urea) conditions, the appearance of these bands is surprising. It is interesting to note that this signal is not seen with RBN010860 treatment (Fig. S5A, HA-ubiquitin blot, 15-50kDa range) and these bands appear slightly bigger in the NUDT16, MD1 and hydroxylamine samples than in the TssM* samples (Fig. 2B, HA-ubiquitin blot, 15-50kDa range). The authors should discuss possible explanations for the origin of these bands.”

Response: We believe that the lower MW bands are released Ub from the MARUbe species, likely due to boiling the samples to elute GFP-PARP10 off the beads. Indeed, a recent paper from the Matic group demonstrated that the ester bond in an ester-linked Ub-ADPr peptide is thermally unstable [Longarini and Matic, Nature Communications 15; 4239 (2024)]. We hypothesize that the MW shift of these lower MW Ub bands in NUDT16/MD1/NH₂OH treatments compared to TssM is due to the added mass of AMP (NUDT16 treatment) or ADPr (MD1/NH₂OH). Given that the NH₂OH-released Ub bands are still mass-shifted could suggest NH₂OH may not have removed the ester-linked Ub on the adenine-proximal ribose. Another possibility is that the resultant hydroxamic acid Ub is sufficient to slow down the migration of Ub species. The RBN010860 treatment supports these results since this prevents MARUbylation and, therefore, less MARUbe is released from boiling.*

4. “There is no data or indeed any discussion of the E3 ligases that could catalyse the isopeptide-linked ubiquitination or the MAR-linked K11 poly-Ub chain modification on PARP10. Ideally, the authors could provide data testing the effect of depleting some candidate E3 ligases, but failing that, at least a discussion of some candidates is necessary.”

Response: We tried knocking down DTX2, but the siRNA KD did not work efficiently despite multiple attempts. We also purchased DTX2 KO cells; however, upon further validation, they were not actual KO cells. We agree with the referee that including a discussion of potential E3 ligases in the paper's discussion section would be important, and we have now added this to the Discussion section of the new manuscript.

5. “Can the authors demonstrate that depleting Cezanne, the K11 poly-Ub deubiquitinase identified in the UbiCREST assay, or any other K11 deubiquitinase, affects PARP10 modification or PARP10 protein levels?”

Response: It would be convenient if the DUB we identified in the UbiCREST assay is the same DUB that acts on PARP10 in cells. However, this may not be the case, and if we see no effect on the knockdown of Cezanne or the various K11 DUBs, a non-specific DUB can also remove the K11 chains. We agree with the referee that identifying the K11 DUB (as well as the MARUbe DUB) is a critical next step, and we plan to take an unbiased screening approach for these

efforts; however, we believe that these studies are outside the scope of this manuscript.

Minor concerns:

6. “Is the PARP10 auto-MARylation signal the only ADPr signal induced with GFP-PARP10 expression? An image of a full molecular weight range ADPr blot would resolve this.”

Response: Yes. All uncropped blots have been provided.

7. “Can ADPr detection reagents detect an ADPr moiety on the released di-Ub species in the IP supernatants (Fig. S3 and 4A)?”

Response: The referee suggests a good idea. In Fig. 4A, NUDT16 cleaves ADPr at the phosphodiester, leaving the phospho-ribose fragment that, unfortunately, is not detectable by any ADPr antibody. We did try to detect the released Ub species from macrodomain treatment (as shown in previous Fig. S3) with the poly/mono ADPr antibody but did not observe a signal. This could be due to the antibody’s poor ability to bind free ADPr vs. canonical ADP-ribosylation (e.g., glu/asp linked).

8. “There are additional bands in the 50-75 kDa range detected using the Ub(FK2) antibody in Fig. 2B, only in the TssM* sample. Can the authors determine the origin of these species?”

Response: This is an astute observation by the reviewer. There are two possible reasons that may explain the observed increase in auto-Ub of DTX2 in the TssM sample. One, we think that DTX2 can ubiquitylate ATP on the hydroxyl ribose. If so, then even after desalting the reaction, TssM* can remove the Ub from the unused ATP-Ub, and therefore, the free ATP can be used for additional DTX2 auto-polyUb (isopeptide-linked). An alternative explanation is that TssM* may bind but not cleave the DTX2 auto-poly Ub, thereby protecting and stabilizing it during the reaction.*

As a revision experiment, we stringently purified the E2~Ub complex to test the first hypothesis using column chromatography. Using this purified E2~Ub complex, we no longer observe these “extra” Ub bands in the TssM lane (new Fig. 2B). As expected, the pattern of the 37-50 kDa DTX2-(Ub)_n bands look similar for the buffer and TssM* lanes.*

9. “The blots in Fig. S5 are substantially spliced, making it difficult to properly compare signal intensities between samples. Based on other data in the manuscript (Fig. 3B), wouldn’t the authors expect RBN010860 to reduce the GFP-PARP10 ubiquitination to around ~60% of the DMSO control?”

Response: Full blots are now provided, making it clear that the blots were imaged at the same exposure. One explanation for the lack of reduction of Ub from inhibitor treatment in cells is because inhibition of MARUbylation leads to more canonical ubiquitylation.

10. “Fig. 4C shows that Cezanne is the only DUB within the panel that can hydrolyse K11 linkages, but this experiment alone is insufficient to demonstrate that Cezanne is strictly specific for K11, as other linkages were not tested. I suggest revising the text accordingly.”

Response: The remarkable specificity of Cezanne for K11-linked Ub has been confirmed in multiple other publications [PMID: 29973362, 23827681, 25633630, 27732584]. Regardless, if it were a secondary activity of Cezanne, we would have seen that with the other DUBs tested (e.g., K63 DUB). We added the references demonstrating the specificity of Cezanne for K11 linkages.

Errors/typos:

pg. 5-bottom: "doxycycline (dox)-inducible HEK 293 cells" is imprecise. It is the GFP-PARP10 expression that is dox-induced, not the cells. There are other similar examples, I suggest careful revision.

Fig 2B: indicate the likely TssM bands next to the picture of the coomassie-stained gel

Pg.8: correct the sentence to: "DTX2 was also reported to catalyze isopeptide-linked ..."

The aforementioned typos and grammar have been addressed.

Referee #3:

"The manuscript describes an investigation into the molecular action of PARPs and their ability to transfer ADP-ribose onto protein substrates. The study is mostly focused on PARP10 and its ability to auto-modify itself. Evidence is provided that the ADP-ribose moiety is further modified with poly-ubiquitin chains, though it remains possible that ubiquitylation occurs during lysate preparation and not within the normal context of cellular function. PARP10 was also shown to be directly modified with poly-ubiquitin chains (eg. auto-ubiquitylation), though the significance of this was unclear. Chain linkage analysis revealed that ubiquitin chains attached to ADP-ribose were formed through Lys 11. Experiments were performed in the presence of MG-132, a well-known proteasome inhibitor, or TAK-243, an inhibitor of ubiquitylation. MG-132 treatment resulted in the appearance of a band migrating above GFP-PARP10 and apparent stabilization of the ADP-ribosylated form (but no apparent changes in the levels of unmodified GFP-PARP10). Cycloheximide chase experiments demonstrated that GFP-PARP10 protein levels are unstable in a manner that depended on PARP10 enzymatic activity.

Taken together, the findings are convincing that a poly-ubiquitin chain may be established onto the ADP-ribose moiety on auto-modified GFP-PARP10. However, the physiological relevance of this modification is dubious. While changes occur to GFP-PARP10 upon treatment with proteasome inhibitor, there is no clear effect on GFP-PARP10 levels (Figure 5A). Furthermore, the degree of modification on GFP-PARP10 appears to be very subtle (Figure 3B). While this latter point was acknowledged by the authors in the Discussion section (and attention was drawn to the fact that ester bonds are labile), this still doesn't explain the lack of PARP10 stabilization upon treatment of cells with MG-132. As such, in the reviewer's opinion, the authors convincingly demonstrate that while dual modification of PARP10 by ADP-ribose and a poly-ubiquitin chain is possible in principle, it is still very much an open question whether this is somehow an artifact of the experiment or of bona fide biological relevance."

Response: We appreciate the referee's thoughtful comments and suggestions, which motivated us to delve deeper into the physiological significance of MARUbylation. We sought to determine if we could detect endogenous MARUbylation by PARP10 and other MARYlating PARPs. As detailed in our response to referee #2's first comment, we found that treatment of HEK 293 cells with the type I interferon, IFN- β , induced endogenous expression of PARP10 (as well as PARP14) and a MARYlation smear originating at the molecular weight of PARP10 (new Fig. 5A). PARP inhibitor studies and treatment with TssM and TssM provide compelling evidence that this smear represented endogenous MARUbylation on PARP10 and PARP14 (new Fig. 5B, F-H). We believe these results demonstrate genuine biological relevance for MARUbylation.*

The referee also highlights an important point about the absence of an apparent effect on GFP-PARP10 levels following MG132 treatment. We repeated the GFP-PARP10 stability experiment using MG132 and another proteasome inhibitor, bortezomib. Although we did not consistently observe an increase in a distinct upper GFP-PARP10 band, we repeatedly

observed a high (>150 kDa) molecular weight (MW) GFP-PARP10 and corresponding MARYlation smear, following treatment with MG132 for progressively longer durations (new **Fig. EV5B**). In light of the lack of a clear effect of proteasome inhibition on GFP levels, we felt it most appropriate to omit the claim (and related data) in the revised version of our manuscript that MARUbylation of PARP10 leads to its degradation. The underlying cause of the increased MARUbylation upon proteasomal inhibition is unclear. One possibility is that proteasome inhibition stabilizes a repressor of MARUbylation removal (e.g., an ADPr hydrolase or DUB). We discuss this possibility in the revised manuscript.

Finally, we investigated whether MARUbylation occurs in lysates or inside the cell. We compared lysing dox-inducible GFP-PARP10 cells or IFN- β -stimulated cells in buffer containing either 1% TX-100 (as done before) or 1% SDS (denaturing conditions). Lysing cells in 1% SDS more effectively preserved the MARYlation smear, which could be reduced by treatment with TssM and TssM* (new **Fig. 5B-F**). These results support the notion that MARUbylation is indeed occurring in cells.

Comments:

1. "Page 6, last line. The authors claim that susceptibility of the sample to hydroxylamine is evidence that MARYlation occurs on Glu/Asp residues. How can they rule out that the modification is not occurring on Ser residues, since oxyester linkages to Ser may also be hydrolyzed by hydroxylamine? It was also stated that "we treated lysates with neutral hydroxylamine", but this is likely not accurate since the pKa of hydroxylamine is approximately 6."

Response: We can rule out Ser residues because MARYlation of Ser would generate an ether, not an oxyester linkage (Fig. 3R below). As such, Ser-ADPr is not cleaved by hydroxylamine [Palazzo et al., eLife 7;34334 (2018)]. We diluted NH₂OH in a pH 7.5 HEPES buffer, and the resulting pH was 7.5. We see how saying the word neutral can be misleading. We have changed the text as follows: "Therefore, we treated lysates with hydroxylamine (buffered at pH 7.5), a chemical that can rapidly cleave ester-linked ADPr".

Fig. 3R. Chemical structures of Asp/Glu-ADPr (top) and Ser-ADPr (bottom).

2. "Figure 2B. The authors claim that DTX2 auto-ubiquitylation occurs on Lys residues which appears consistent with the anti-Ubiquitin FK2 Ab Western blot. Laddering is observed when

DTX is added to the reaction but then disappears upon treatment with the DUB TssM. What's unclear is why the laddering is more pronounced and the intensity of the bands so much greater in the sample treated with TssM*, a mutant variant that is supposedly far more specific as an esterase compared with isopeptidase activity.”

Response: The referee brings up a good point that was also brought up by another referee. There are two possible reasons that may explain the observed increase in auto-Ub of DTX2 in the TssM sample. One, we think that DTX2 can ubiquitylate ATP on the hydroxyl ribose. If so, then even after desalting the reaction, TssM* can remove the Ub from the unused ATP-Ub, and therefore, the free ATP can be used for additional DTX2 auto-polyUb (isopeptide-linked). An alternative explanation is that TssM* may bind but not cleave the DTX2 auto-poly Ub, thereby protecting and stabilizing it during the reaction.*

As a revision experiment, we stringently purified the E2~Ub complex to test the first hypothesis using column chromatography. Using this purified E2~Ub complex, we no longer observe these “extra” Ub bands in the TssM lane (new **Fig. 2B**). As expected, the pattern of the 37-50 kDa DTX2-(Ub)_n bands look similar for the buffer and TssM* lanes.*

3. “Figure 3B. There appears to be no negative control for the immuno-precipitation such as GFP alone. This significantly weakens the interpretations made by the authors.”

*Response: Stringent (denaturing) washing conditions are used to remove any possible background signal. In **Fig. EV4D**, the ADPr signal is eliminated with an inhibitor of PARP10. Moreover, in **Fig. EV4F**, we enriched a different GFP-tagged PARP (i.e., GFP-PARP7), and the pattern of ADPr/HA-Ubiquitin is entirely different from the GFP-PARP10. We believe these results indicate that the ADPr/HA-Ubiquitin signals observed in the pulldowns with GFP-trap beads arise from the expressed GFP-tagged PARPs.*

Prof. Michael S Cohen
Oregon Health and Science University, Portland, OR, USA
Chemical Physiology and Biochemistry

24th Jan 2025

Re: EMBOJ-2024-118401R
Ubiquitin is directly linked via an ester to protein-conjugated mono-ADP-ribose

Dear Dr. Cohen,

Thank you for submitting your revised manuscript for our consideration. All original reviewers have now looked at it once more, and I am pleased to say that they were broadly satisfied with your responses and revisions. After incorporation of minor textual modifications requested by referees 2 and 3, we shall therefore be happy to accept your manuscript for EMBO Journal publication.

In addition, there also remain several editorial issues that should still be addressed at this point:

- Most importantly, we still need you to complete and upload the Source Data Checklist that had been sent to you by our Source Data curator, Hannah Sonntag (I am attaching it once more to this message). Please also double-check to make sure that all requested Source Data items have been uploaded.
Also, please note that Source data files need to be saved in a scheme one figure per one folder, and then uploaded as .zip files. E.g. all the Source data files for figure 1 need to be saved in a single folder and this needs to be zipped and then uploaded as "SD figure 1.zip" file. For EV and/or appendix figures, on the other hand, ZIP together all source data in one single archive.
- Please adjust the format of the reference list and of the in-text citations according to EMBO Journal format (alphabetical order, author name et al + year, up to 10 authors listed before abbreviating with et al in the bibliography...)
- On the abstract page of the manuscript, please include 4-5 general keyword terms to enhance searchability.
- Please rename the Conflict of Interest section into "Disclosure and Competing Interests Statement", in accordance with our updated Guide to Authors (<https://www.embopress.org/competing-interests>)
- As we are switching from a free-text author contribution statement towards a more formal statement based on Contributor Role Taxonomy (CRediT) terms, please remove the present Author Contribution section and instead specify each author's contribution(s) directly in the Author Information page of our submission system during upload of the final manuscript. See <https://casrai.org/credit/> for more information.
- Please double-check to make sure to all relevant funding information in the manuscript is congruent with the info entered into our submission system (currently missing in the submission system but cited in the text: Achievement Rewards for College Scientists (ARCS), National Cancer Institute NRSA F31 fellowship F31CA284712)
- Please remove the Reagents & Tool tables from the Methods section, and instead upload it separately - making sure to use the template file downloadable from our Guide to Authors at <https://www.embopress.org/page/journal/14693178/authorguide#structuredmethods>
- Please provide suggestions for a short 'blurb' text prefacing and summing up the conceptual aspect of the study in two sentences (max. 250 characters), followed by 3-5 one-sentence 'bullet points' with brief factual statements of key results of the paper; they will form the basis of an editor-written 'Synopsis' accompanying the online version of the article. Please also upload a synopsis image, which can be used as a "visual title" for the synopsis section of your paper. The image should be in PNG or JPG format, and please make sure that it remains in the modest dimensions of (EXACTLY) 550 PIXELS WIDE and 300-600 PIXELS HIGH.
- Finally, during routine pre-acceptance checks, our data editors have raised the following queries regarding figures, data, and legends, which I would ask you to address (ideally using the Track Changes option):
 1. Please note that the exact p values are not provided in the legends of figures 3C, D
 2. Please indicate what */ **/ ***/ **** represents; if this represents p value(s), please indicate the statistical test used and where appropriate, specify the exact p value in the legend(s) of figure(s) EV 4C

I am therefore returning the manuscript to you for a final round of revision, to allow you to make these modifications and upload the revised files. Once we will have received them, we should be ready to swiftly proceed with formal acceptance and production

of the manuscript.

With kind regards,

Hartmut

- 1) Every manuscript requires a Data Availability section (even if only stating that no deposited datasets are included). Primary datasets or computer code produced in the current study have to be deposited in appropriate public repositories prior to resubmission, and reviewer access details provided in case that public access is not yet allowed. Further information: embopress.org/page/journal/14602075/authorguide#dataavailability
- 2) Each figure legend must specify
 - size of the scale bars that are mandatory for all micrograph panels
 - the statistical test used to generate error bars and P-values
 - the type error bars (e.g., S.E.M., S.D.)
 - the number (n) and nature (biological or technical replicate) of independent experiments underlying each data point
 - Figures may not include error bars for experiments with $n < 3$; scatter plots showing individual data points should be used instead.
- 3) Revised manuscript text (including main tables, and figure legends for main and EV figures) has to be submitted as editable text file (e.g., .docx format). We encourage highlighting of changes (e.g., via text color) for the referees' reference.
- 4) Each main and each Expanded View (EV) figure should be uploaded as individual production-quality files (preferably in .eps, .tif, .jpg formats). For suggestions on figure preparation/layout, please refer to our Figure Preparation Guidelines: <http://bit.ly/EMBOPressFigurePreparationGuideline>
- 5) Point-by-point response letters should include the original referee comments in full together with your detailed responses to them (and to specific editor requests if applicable), and also be uploaded as editable (e.g., .docx) text files.
- 6) Please complete our Author Checklist, and make sure that information entered into the checklist is also reflected in the manuscript; the checklist will be available to readers as part of the Review Process File. A download link is found at the top of our Guide to Authors: embopress.org/page/journal/14602075/authorguide
- 7) All authors listed as (co-)corresponding need to deposit, in their respective author profiles in our submission system, a unique ORCID identifier linked to their name. Please see our Guide to Authors for detailed instructions.
- 8) Please note that supplementary information at EMBO Press has been superseded by the 'Expanded View' for inclusion of additional figures, tables, movies or datasets; with up to five EV Figures being typeset and directly accessible in the HTML version of the article. For details and guidance, please refer to: embopress.org/page/journal/14602075/authorguide#expandedview
- 9) To facilitate reproducibility and cross-laboratory adoption of methodologies, please structure the Materials & Methods section as outlined in our guide to authors, including a completed Reagents and Tools Table that can be downloaded from our author guidelines as well (<https://www.embopress.org/page/journal/14602075/authorguide#structuredmethods>).
- 10) Digital image enhancement is acceptable practice, as long as it accurately represents the original data and conforms to community standards. If a figure has been subjected to significant electronic manipulation, this must be clearly noted in the figure legend and/or the 'Materials and Methods' section. The editors reserve the right to request original versions of figures and the original images that were used to assemble the figure. Finally, we generally encourage uploading of numerical as well as gel/blot image source data; for details see: embopress.org/page/journal/14602075/authorguide#sourcedata

At EMBO Press, we ask authors to provide source data for the main manuscript figures. Our source data coordinator will contact you to discuss which figure panels we would need source data for and will also provide you with helpful tips on how to upload

and organize the files.

Further information is available in our Guide For Authors:

In the interest of ensuring the conceptual advance provided by the work, we recommend submitting a revision within 3 months (24th Apr 2025). Please discuss the revision progress ahead of this time with the editor if you require more time to complete the revisions. Use the link below to submit your revision:

Link Not Available

Referee #1:

The authors have performed several experiments and now provide additional data to support the existence of ester-linked MARubylation in cells. Although they were unable to confirm with statistical support that MARubylation leads to destabilization and proteasomal degradation of PARP10, they did manage to show that interferon beta treatment induces PARP10 and PARP14 in cells and generates strong MARylation signal that collapses upon TssM treatment of lysates, providing compelling support for endogenous MARubylation induced by innate immunity. This discovery is timely and important as recent work from other labs already identified E3 ubiquitin ligases able to carry out this modification in vitro. What the roles of this dual modification in cells and its impact on targeted proteins are will be interesting grounds for future research.

Referee #2:

Overall, I find the revised version to address my comments to satisfaction. However, I would suggest minor changes to the text, as described below:

1. The minor shift in the migration pattern of NAD⁺-Ub/ADPr-Ub relative to Ub in NUDT16-treated samples in Fig. 2C (which could be the result of various ADPr remnants attached to Ub), together with the surprising data in Fig. 2R showing loss of desthiobiotin after NUDT16 incubation of DTB-NAD-Ub, are insufficient to conclusively demonstrate that the Ub modification is indeed on the 3'OH of the A-ribose of ADPr, such that this conclusion is largely based on previous work (ref. 18). Therefore, I suggest rewording sentences in which the A-ribose 3'OH attachment is mentioned, to leave the possibility of alternative ADPr-Ub linkages open, and discussing this in the text.
2. The new data using IFN treatment is an important advance to demonstrate that MARubylation is physiologically relevant. However, the evidence that the signals detected by western blotting are indeed modification of PARP10 and PARP14 is based solely on the molecular weight of the observed bands, which is circumstantial. Unless the authors can demonstrate by immunoprecipitation or other methods that PARP10 and PARP14 are indeed the target proteins, the interpretation of data in Fig. 5 should not exclude the possibility of other targets.
3. As stated by the authors in their response, the new experiments using IFN treatment were inspired by Kar et al EMBO 2024 (and presumably also by the accompanying Ribeiro et al EMBO 2024). Therefore, both papers should be cited in the manuscript.

Referee #3:

The revised manuscript by Bejan and colleagues is now a pleasure to read with clear and concise statements supported by the data. I fully support publishing the work in EMBO journal with a few minor suggestions (left to the discretion of the authors and editor).

P13. These results strongly indicate that IFN- β can drive endogenous MARubylation, predominately occurring on PARP14 and PARP10.

While the reviewer agrees that the single prominent band in figure 5F (top panel) may be PARP10, it is still possible that the band may represent a different protein. This is acknowledged by the authors in the Discussion section, so why not change the sentence to "...MARubylation, predominantly occurring on PARP14 and also likely on PARP10 as well."

P15. Lastly, Ub can also undergo acetylation (on Lys) or phosphorylation (on Ser/Thr/Tyr).

This statement seems out of place considering the counter point in the 2nd sentence with the topic sentence of the paragraph. Specifically, that there are cases known where residues are modified with two types of PTMs. Also, in the 2nd sentence, it would be more appropriate to refer to the modified species as a residue and not an amino acid.

Finally, while this reviewer fully endorses noncanonical scientific writing in the literature, the final sentence of the Discussion may be just a bit too cute, even for me. I would consider a bit of editing that maintains the spirit but is befitting of the seriousness of the work.

Manuscript number: EMBOJ-2024-118401

Response to 2nd Round Reviewer Comments

Referee #1:

The authors have performed several experiments and now provide additional data to support the existence of ester-linked MARubylation in cells. Although they were unable to confirm with statistical support that MARubylation leads to destabilization and proteasomal degradation of PARP10, they did manage to show that interferon beta treatment induces PARP10 and PARP14 in cells and generates strong MARylation signal that collapses upon TssM treatment of lysates, providing compelling support for endogenous MARubylation induced by innate immunity. This discovery is timely and important as recent work from other labs already identified E3 ubiquitin ligases able to carry out this modification in vitro. What the roles of this dual modification in cells and its impact on targeted proteins are will be interesting grounds for future research.

Response: We thank the referee for their support of our revised manuscript.

Referee #2:

Overall, I find the revised version to address my comments to satisfaction. However, I would suggest minor changes to the text, as described below:

Response: We are glad the referee is satisfied with our revised manuscript.

1. The minor shift in the migration pattern of NAD⁺-Ub/ADPr-Ub relative to Ub in NUDT16-treated samples in Fig. 2C (which could be the result of various ADPr remnants attached to Ub), together with the surprising data in Fig. 2R showing loss of desthiobiotin after NUDT16 incubation of DTB-NAD-Ub, are insufficient to conclusively demonstrate that the Ub modification is indeed on the 3'OH of the A-ribose of ADPr, such that this conclusion is largely based on previous work (ref. 18). Therefore, I suggest rewording sentences in which the A-ribose 3'OH attachment is mentioned, to leave the possibility of alternative ADPr-Ub linkages open, and discussing this in the text.

Response: The referee raises a fair point. We have removed claims of Ub occurring on the 3'-OH of the A-ribose in regard to Fig 2C, and discussed the possibility of other attachment points in the text:

“However, we can't exclude the possibility of Ub attachment occurring on a hydroxyl group of the distal ribose of NAD⁺ or ADPr.”

2. The new data using IFN treatment is an important advance to demonstrate that MARubylation is physiologically relevant. However, the evidence that the signals detected by western blotting are indeed modification of PARP10 and PARP14 is based solely on the molecular weight of the observed bands, which is circumstantial. Unless the authors can demonstrate by immunoprecipitation or other methods that PARP10 and PARP14 are indeed the target proteins, the interpretation of data in Fig. 5 should not exclude the possibility of other targets.

Response: The referee brings up a good point. Identifying the exact targets of MARUbylation upon IFN- β is important and will be the goal of future studies. Referee #3 brought up a similar point, and we have adjusted the text to be more speculative on whether PARP10 itself is MARUbylated:

“These results strongly indicate that IFN- β can drive endogenous MARUbylation, likely occurring on PARP14 and PARP10; however, we cannot exclude the possibility that IFN- β -induced MARUbylation occurs on other proteins, perhaps targets of PARP14 and PARP10.”

3. As stated by the authors in their response, the new experiments using IFN treatment were inspired by Kar et al EMBO 2024 (and presumably also by the accompanying Ribeiro et al EMBO 2024). Therefore, both papers should be cited in the manuscript.

Response: We thank the reviewer for bringing this to our attention. We have cited both papers in the text as follows:

“Additionally, it has recently been shown that PARP14 expression and catalytic activity (MARylation) is strongly induced upon IFN- γ treatment (Kar et al, 2024; Ribeiro et al, 2024). Therefore, we wondered whether IFN- β treatment would induce PARP10 expression and allow for the detection of endogenous MARUbylation”

Referee #3:

The revised manuscript by Bejan and colleagues is now a pleasure to read with clear and concise statements supported by the data. I fully support publishing the work in EMBO journal with a few minor suggestions (left to the discretion of the authors and editor).

Response: We are thrilled the referee finds our revised manuscript a pleasure to read.

P13. These results strongly indicate that IFN- β can drive endogenous MARUbylation, predominately occurring on PARP14 and PARP10.

While the reviewer agrees that the single prominent band in figure 5F (top panel) may be PARP10, it is still possible that the band may represent a different protein. This is acknowledged by the authors in the Discussion section, so why not change the sentence to "...MARUbylation, predominantly occurring on PARP14 and also likely on PARP10 as well."

Response: This is a good point, that referee #2 also raised. We have modified the sentence as referee #3 suggested.

P15. Lastly, Ub can also undergo acetylation (on Lys) or phosphorylation (on Ser/Thr/Tyr).

This statement seems out of place considering the counter point in the 2nd sentence with the topic sentence of the paragraph. Specifically, that there are cases known where residues are modified with two types of PTMs. Also, in the 2nd sentence, it would be more appropriate to refer to the modified species as a residue and not an amino acid.

Response: We see the confusion the referee raises regarding the statement about Ub undergoing acetylation or phosphorylation (on different residues). The point we were trying to

make is that this is still a double PTM (Ubiquitylation being the first PTM, and acetylation or phosphorylation being the second PTM). We have reworded the text to make that clearer:

“Lastly, Ub on ubiquitylated targets can subsequently undergo acetylation on lysine (Lys) or phosphorylation on serine, threonine, or tyrosine (Ser/Thr/Tyr) (Swatek & Komander, 2016)”

We have also replaced the word “amino acid” to “residue” in the 2nd sentence, as suggested.

Finally, while this reviewer fully endorses noncanonical scientific writing in the literature, the final sentence of the Discussion may be just a bit too cute, even for me. I would consider a bit of editing that maintains the spirit but is befitting of the seriousness of the work.

Response: While we understand maintaining a serious tone in scientific writing, the final sentence is a fun way to end the paper, and we believe readers will enjoy the added voice and character it provides to the story.

Prof. Michael S Cohen
Oregon Health and Science University, Portland, OR, USA
Chemical Physiology and Biochemistry

7th Feb 2025

Re: EMBOJ-2024-118401R1
Ubiquitin is directly linked via an ester to protein-conjugated mono-ADP-ribose

Dear Mike,

Thank you for submitting your final revised manuscript for our consideration. I am pleased to inform you that we have now accepted it for publication in The EMBO Journal.

With kind regards,

Hartmut
